# Phenotypic characteristics of peripheral immune cells of Myalgic encephalomyelitis/ chronic fatigue syndrome via transmission electron microscopy: A pilot study

Fereshteh Jahanbani[1]*, Rajan D. Maynard[1], Justin Cyril Sing[2], Shaghayegh Jahanbani[3], John J. Perrino[4], Damek V. Spacek[5], Ronald W. Davis[6], Michael P. Snyder[1]*

1 Department of Genetics, Stanford University School of Medicine, Stanford, California, United States of America, 2 Department of Molecular Genetics, Donnelly Centre, University of Toronto, Toronto, Ontario, Canada, 3 Division of Immunology and Rheumatology, Stanford University School of Medicine, and VA Palo Alto Health Care System, Palo Alto, California, United States of America, 4 Stanford Cell Sciences Imaging Facility (CSIF), Stanford University School of Medicine Stanford, Stanford, California, United States of America, 5 Karius Incorporated, Redwood City, California, United States of America, 6 ME/CFS Collaborative Research Center at Stanford, Stanford Genome Technology Center, Stanford University School of Medicine, Palo Alto, California, United States of America

☯ These authors contributed equally to this work.
* fjahania@stanford.edu (FJ); mpsnyder@stanford.edu (MPS)

**Data Availability Statement:** TEM images for this study are publicly available from the Stanford Digital Repository (https://purl.stanford.edu/

## Abstract

Myalgic encephalomyelitis/chronic fatigue syndrome (ME/CFS) is a complex chronic multi-systemic disease characterized by extreme fatigue that is not improved by rest, and worsens after exertion, whether physical or mental. Previous studies have shown ME/CFS-associated alterations in the immune system and mitochondria. We used transmission electron microscopy (TEM) to investigate the morphology and ultrastructure of unstimulated and stimulated ME/CFS immune cells and their intracellular organelles, including mitochondria. PBMCs from four participants were studied: a pair of identical twins discordant for moderate ME/CFS, as well as two age- and gender- matched unrelated subjects—one with an extremely severe form of ME/CFS and the other healthy. TEM analysis of CD3/CD28-stimulated T cells suggested a significant increase in the levels of apoptotic and necrotic cell death in T cells from ME/CFS patients (over 2-fold). Stimulated Tcells of ME/CFS patients also had higher numbers of swollen mitochondria. We also found a large increase in intracellular giant lipid droplet-like organelles in the stimulated PBMCs from the extremely severe ME/CFS patient potentially indicative of a lipid storage disorder. Lastly, we observed a slight increase in platelet aggregation in stimulated cells, suggestive of a possible role of platelet activity in ME/CFS pathophysiology and disease severity. These results indicate extensive morphological alterations in the cellular and mitochondrial phenotypes of ME/CFS patients' immune cells and suggest new insights into ME/CFS biology.

zm622tr7008). The de-identified FASTQ files for 4 whole exome sequencing for this study are also publicly available from the Stanford Digital Repository (https://purl.stanford.edu/jd768nw9509).

**Funding:** Michael P. Snyder received from the National Institute of Health (NIH) 5RM1HG00773508 Ronald W. Davis, Fereshteh Jahanbani, and Rajan D. Maynard received Open Medicine Foundation (OMF) funding (https://www.omf.ngo/extended-big-data-study/). Fereshteh Jahanbani received a funding from an anonymous private donor.

**Competing interests:** The authors declare that they have no known competing financial interests or personal relationships that could have appeared to influence the work reported in this paper.

## Introduction

Myalgic encephalomyelitis/chronic fatigue syndrome (ME/CFS) is a complex, chronic, debilitating, multi-systemic disease with many comorbidities. It is characterized by chronic idiopathic fatigue, post-exertional malaise (PEM), sleep problems, cognitive impairments/brain fog, and/or orthostatic intolerance [1, 2]. ME/CFS is associated with substantial reductions in previous levels of occupation, education, and social functioning. The progression of symptoms can lead to severe physical disability and a significant reduction in quality of life [3, 4], which is similar to that observed in patients with long COVID [5, 6]. The remarkable similarities between long COVID sequelae and ME/CFS has garnered considerable attention [7]. A better understanding of ME/CFS etiology and pathogenesis may contribute to improved understanding of this general class of chronic illness.

It is estimated that between 836,000 to 2.5 million people within the USA suffer from ME/CFS but the actual number might be higher as a significant percentage of patients do not get the proper diagnosis [8]. Moreover, given the likelihood that a number of COVID19 patients might develop ME/CFS related symptoms, the total number of ME/CFS patients worldwide may increase dramatically within a short period of time. Due to its chronic nature, ME/CFS has significant economic burdens per year in medical bills and lost income from patients, family, and caregivers [9–11]. However, few studies have examined the direct and indirect costs of ME/CFS. In 2008 the economic impacts of ME/CFS in USA was estimated to be $17 to $24 billion dollars [11]. Considering ME/CFS high prevalence rates and a potential link between post-COVID conditions (such as long COVID) and new cases of ME/CFS, there is an immediate need to reevaluate the prevalence rates and global economic impacts of ME/CFS at the individual, family, and societal level [12, 13]. In the absence of specific diagnostic tests, however, it is difficult to precisely determine the disease burden and prevalence [14]. A 2021 paper estimates a rough doubling in both ME/CFS prevalence and economic impact to 36–51 billion dollars per year [15]. In a subsequent paper, the same authors estimate a significant increase in both ME/CFS prevalence (between five and nine million COVID-related or unrelated-ME/CFS cases) and economic impact ($258 billion to $362 billion for direct and indirect costs) [7].

While the etiology and pathogenesis of ME/CFS are still unknown, a growing body of evidence implicates neuro-immune-metabolic-endocrine-microbiome circuit dysregulation as an underlying feature of ME/CFS [16]. Several studies have shown alterations of immune cell function in patients with ME/CFS, including changes in number and function of T cells, B cells and natural killer (NK) cells, as well as alterations in cytokine production and chromatin landscape [17–21]. Many researchers have proposed that metabolic impairment and mitochondrial aberrations are also implicated in ME/CFS pathophysiology [22–28]. Mitochondria, as the "powerhouses" of the cell because of their critical role in energy production, play a key role in innate and adaptive immune system responses, helping to resolve inflammation and to maintain homeostasis [29, 30]. Mitochondrial dysfunction contributes to the pathogenesis and progression of numerous human diseases, including cancer, neurodegenerative and cardiovascular disorders, and traumatic brain injury. Mitochondrial function also coordinates cell survival and cell death, including programmed (apoptosis, necroptosis, pyroptosis, ferroptosis, and autophagy) and non-programmed (necrosis) cell death [31].

Several studies point to mitochondrial dysfunction as a key contributor to an array of ME/CFS symptoms, including muscle weakness, pain, cognitive decline, and the dynamics of these symptoms [32]. Other studies have shown that cellular bioenergetics such as basal respiration, ATP production, maximal respiration, and reserve capacity are impaired in patients with ME/

CFS [33]. Mitochondrial bioenergetics are closely linked to mitochondrial structure. Mitochondria have intricate mechanisms that allow them to change size, shape, and position over the course of a few seconds and to undergo a fission or fusion [34]. These alterations in mitochondrial morphology impact bioenergetics; conversely changes in bioenergetics often cause morphological alterations and both appear to be regulated by surrounding cues crucial to cell health [35]. For example, elevated levels of reactive oxygen species (ROS) due to oxidative stress can cause mitochondrial fragmentation, hypoxia can cause fragmented and donut shaped mitochondria, and alterations in mitochondrial shape, size and cristae conformation can indicate active or inactive mitochondria [36, 37].

While emerging evidence points to oxidative stress, aberrant immune responses, and mitochondrial dysregulation in the pathogenesis of ME/CFS, there are limited studies on morphological changes in immune cells, mitochondria, and other cell organelles in ME/CFS patients. One of the best tools to study morphological and ultrastructural changes is electron microscopy. At present only a few such studies of ME/CFS patients have been conducted, and these have mostly examined mitochondrial abnormalities in muscle cells although one paper analyzed frozen blood cells [38–40]. Considering the multi-systemic condition of the disease and its underlying immune dysregulation, we hypothesized that immune cells and their organelles' ultrastructure might be altered in ME/CFS patients. To this end, we analyzed peripheral blood mononuclear cells (PBMCs) from a monozygotic twin pair discordant for a moderate form of ME/CFS and another pair of age- and gender-matched unrelated subjects, one with extremely severe ME/CFS and the other healthy.

PBMCs are key components of the body's immune system and are characterized by a single rounded nucleus and consist of heterogeneous cell population comprising lymphocytes (B cells (~15%), T cells (~70%), monocytes (~5%), natural killer (NK) cells (~10%)) [41], dendritic cells, and monocytes. PBMCs can provide a snapshot of the body's circulating immune compartment and are frequently used in immunological studies, vaccine development, drug, biomarker, and toxicity screening and discovery [42–48]. We used Transmission Electron Microscopy (TEM) to study the phenotypic characteristics of the isolated PBMCs, and the ultrastructure of their organelles such as mitochondria. Our results showed a markedly increased induction in both apoptotic and necrotic cell death in stimulated T cells from ME/CFS patients, which was also associated with disease severity. While there was no significant difference in the number of mitochondria between groups, stimulated T cells from ME/CFS patients exhibited a significant increase in swollen mitochondria, which may be indicative of primary or secondary mitochondrial dysfunction. Interestingly, stimulated T cells from ME/CFS patients showed a significant increase in the number of cells carrying more than 3 swollen or 6 abnormal mitochondria per cell.

Our findings lend further support to the evidence of impairment in energy production in ME/CFS patients, due to mitochondrial dysfunction. We also found that a subset of PBMCs in the extremely severe ME/CFS patient contained large lipid droplet-like vesicles, which could be a contributing factor to the ME/CFS development or a lipid storage disorder comorbidity in this individual. Elevated intracellular lipid droplet-like vesicles have been previously reported in the muscle biopsy of ME/CFS patients using TEM [38]. Integrating TEM analyses with whole exome sequencing data suggested a missense mutation in SMPD1 (sphingomyelin phosphodiesterase 1, acid lysosomal) variant might play a role in the increased lipid droplet-like vacuoles in this extremely severe ME/CFS patient. Finally, we noted an increase in platelet aggregates, which might be associated with some of the symptoms observed in ME/CFS patients that resemble mast cell activation syndrome.

## Methods

### Participants

Participants consisted of male identical twins, discordant for ME/CFS, as well as one extremely severe male ME/CFS patient and age-, and gender-, and BMI-matched healthy participant. Both patients were diagnosed with ME/CFS by two established ME/CFS specialists, based on the 2003 Canadian Consensus Criteria (CCC), 2011 International Consensus Criteria (ICC), and 2015 Institute of Medicine Criteria (IOMC) criteria. The patient age was 31.5±2.1 years and the control age was 30 ±0 years. For the ill cohort, the duration of illness was 7.5±0.7 years. Prior to the onset of illness, both patients were successful professionals. The moderately affected patient is still holding a job, while the extremely severely ill patient has been bed bound for the past two years and is totally dependent on his caregivers. Samples were gathered after informed written consent. Participants' sample ID can be found in Table 1.

### Ethics statement

This study was approved by Stanford Human Research Protection Program Institutional Review Boards (Protocol ID. 40146). Informed written consent was obtained from all participants.

### PBMC isolation

Blood was collected in CPT tubes and PBMCs were isolated following the manufacturer's instructions. Briefly, CPT tubes were centrifuged at 1800 RCF for 15 min. The plasma layer was carefully isolated, and the buffy coat was immediately collected and transferred into a 15 mL conical centrifuge tube. The volume was brought to 15 mL with PBS. The tube was closed and inverted 5 times to mix the mononuclear cells and platelets and centrifuged for 15 min at 1300 RPM. Supernatant was aspirated without disturbing the cell pellet. Cells were washed in PBS for 5 more minutes and collected at 1300 RPM. Isolated PBMCs were subjected to T cell isolation using MACS based sorting.

### T cell isolation

T cells are defined by the expression of CD3 markers on their cell surface, which is associated with T cell receptors, forming TCR/CD3 complexes that play a significant part in the antigen-specific activation of T cells. We used the Pan T cell Isolation Kit (Miltenyi Biotec, Order no. 130-096-535), an immunomagnetic selection-based method, to isolate T cells from the pool of PBMCs. TCRαβ-bearing conventional T cells (conventional CD4+ and CD8+ T cells) were separated from the non-target cells (i.e., monocytes, B cells, PBMC multipotent progenitor stem cell populations, dendritic cells, T cells expressing invariant or semi-invariant TCR chains including NK cells, and the residual granulocytes or erythroid cells) via negative selection. Purified PBMCs were counted and resuspended in 40 μL of buffer per $10^7$ total cells in single-cell suspension. 10 μL of Pan T cell biotin-antibody cocktail was added to PBMC cell

**Table 1. Participants sample IDs, used for the TEM study.**

| Subject | Identification |
| --- | --- |
| Identical male twin with moderate form of ME/CFS | TCFS |
| Identical male twin Healthy control | THC |
| Unrelated male case with extremely severe ME/CFS | UCFS |
| Unrelated age-, gender-, and BMI-matched healthy control | UHC |

**Table 2. Participants' PBMCs subpopulation IDs, used for the TEM study.**

| Cell fraction | Identification |
|---|---|
| T cells | T |
| Stimulated T cells | T+Act |
| PBMC subpopulation lacking T cells | P-T |
| Stimulated PBMC subpopulation lacking T cells | P-T+Act |

suspension, mixed well, and incubated for 5 min at 2–8 ˚C. Non-target cells were magnetically labelled using Pan T Cell microbead cocktail, which contains biotin-conjugated antibodies against CD14, CD15, CD16, CD19, CD34, CD36, CD56, CD123, and CD235a. 30 μL of buffer was added to the mix of biotin-antibody cocktail and PBMCs, followed by the addition of 20 μL of Pan T cell microbead cocktail per $10$ total cells. Cells were mixed 10 times gently and thoroughly and incubated at 10 minutes in the refrigerator (2–8 ˚C), and subjected to subsequent magnetic cell separation following the manufacturer's instructions using a LS column and MACS separator. The cell suspension was applied onto the prewashed LS Column (which was rinsed with 3 mL of provided buffer). Flow-through, which contained unlabeled cells, was kept as an enriched T cells layer (T fraction). The column was further washed with 3 mL of provided buffer and the effluent was added to the previous enriched T cell layer. SL was separated from MACS Separator and washed with a 3 mL buffer. This flow-through was labeled as non-target cells, or PBMC lacking T cell population (P-T fraction) (Miltenyi Biotec, Order number. 130-096-535). Sample ID for different PBMC subpopulations can be found in Table 2.

## T cells and PBMC lacking T cells stimulation

The isolated and purified T cell fraction was suspended in RPMI 1640 medium (Gibco™ 11875085) supplemented with 10% heat-inactivated FBS (fetal bovine serum) (Corning, 35-016-CV), 100 μL/mL penicillin and 100 mg/mL streptomycin. 1x $10^6$ of T cells were incubated at 37˚C in humidified 5% CO2 for 12 hours in the absence or presence of Human anti-CD3 and anti-CD28 monoclonal antibodies covalently linked to superparamagnetic beads (Dynabeads™ Human T-Activator CD3/CD28 for T cell Expansion and Activation Catalog number, Thermo Fisher Scientific, Cat No: 11161D) for T cells stimulation [49]. Similarly, PBMC lacking T cells were also suspended in the cell culture medium mentioned above and cultured at $1 \times 10^6$ density without or with phorbol ester (PMA) at 100 nM (which is used for PBMC stimulation by activating ERK1/2 phosphorylation and p21Cip1/WAF1 pathways) [50]. Stimulated and unstimulated cells were harvested and centrifuged and subjected to fixation for downstream TEM imaging.

## Transmission electron microscopy (TEM)

TEM sample processing and imaging were done at Stanford Cell Sciences Imaging Facility. Samples were fixed in Karnovsky's fixative: 2% Glutaraldehyde (EMS Cat# 16000) and 4% pFormaldehyde (EMS Cat# 15700) in 0.1M Sodium Cacodylate (EMS Cat# 12300) at pH 7.4 for 1 hour, chilled and sent to Stanford's CSIF on ice. After fixation, cells were centrifuged at 4000g for 5 minutes in an Eppendorf MiniSpinPlus, washed in 0.1M sodium cacodylate buffer 3X (pelleting between changes), then mixed with 10% gelatin in 0.1M sodium cacodylate buffer at 35˚C for 10 min and allowed to equilibrate for 5 min. Cells were pelleted again to remove the excess of gelatin, chilled, cut into small pieces, and then processed as below.

The cell pellet in gelatin piece was then allowed to warm to room temperature (RT) in 1% cold osmium tetroxide (EMS Cat# 19100) for 1 hour rotating in a hood, washed 3X with ultra-filtered water, then en bloc stained overnight in 1% uranyl acetate at 4˚C while rotating. Samples were then dehydrated in a series of ethanol washes for 30 minutes, each at 4˚C beginning at 50%, 70%, 95%. Samples were then allowed to rise to RT, and washed with 100% ethanol twice, then incubated with propylene oxide (PO) for 15 min. They were sequentially infiltrated with EMbed-812 resin (EMS Cat#14120), mixed with PO in a ratio of 1:2, 1:1, and 2:1, with 2 hours incubation for each step and finally left in 2:1 ratio of resin to PO for overnight rotating at RT in the hood. The samples were then placed into 100% EMbed-812 for 2–4 hours, and later placed into molds with labels and fresh resin, orientated, and placed into a 65˚ C oven overnight. Ultra-thin sections (roughly 80 nm) were cut, and collected on formvar/Carbon coated 100 mesh copper grids, and stained for 30 seconds in 3.5% uranyl acetate in 50% acetone, which was followed by staining in 0.2% lead citrate for 3 minutes. TEM sections were observed in the JEOL JEM-1400 120kV and photos were taken using a Gatan Orius 4k X 4k digital camera. TEM images are uploaded to Stanford Digital Repository and available via the provided URL link (https://purl.stanford.edu/zm622tr7008).

## Quantitative analyses of TEM images

TEM micrograms were examined to assess morphological characteristics of the PBMC subsets and to semi-quantitatively measure cellular phenotypes such as cell viability, and subcellular organelles' ultrastructure. The number of the cells and the criteria used for the identification of apoptotic or necrotic cells as well as mitochondrial morphological characteristics are provided in the results section.

## Whole exome sequencing

Whole blood was collected into an 8 mL sodium heparinized CPT vacutainer and inverted multiple times. The tube was centrifuged at 1800 g for 15 min at room temperature, resulting in the separation of blood into plasma (top layer), buffy coat containing PBMCs (middle), and erythrocytes, and granulocytes (the lower layer). Plasma was gently aspirated and the PBMCs in the buffy coat were taken out with a 5-mL pipette and added into a 15 mL conical tube. The volume was brought up to 14 mL with Dulbecco's Phosphate-Buffered Saline (PBS, Thermo Scientific). The capped 15 mL conical tube was gently mixed by inversion and centrifuged at 300 g for 15 min at room temperature. The supernatant was carefully aspirated, and the cell pellet was resuspended in 15 mL PBS and centrifuged for 5 min at 300 g at room temperature. The supernatant was again carefully aspirated without disturbing the cell pellet, and the pelleted PBMCs were snap frozen in liquid nitrogen. Genomic DNA was extracted from the frozen PBMC pellets using the AllPrep DNA/RNA Mini Kit (Qiagen, Cat. No. 80404). Whole Exome sequencing was conducted at Personalis Inc. (Menlo Park, CA, USA), using Personalis ACE Clinical Exome sequencing platform. Exome capture was performed using Agilent Sure-Select Clinical Research Exome (SSCR) according to manufacturers' recommendations. Additional supplementation with Personalis ACE proprietary target probes was performed to enhance coverage in difficult to sequence regions within sets of biomedically and medically relevant genes. Details regarding Personalis ACE assay design are described further in Patwardhan et al. 2015 [51]. PBMC specimens were sequenced to an average output of 12 Gb across the 69.4 Mb ACE assay genomic footprint. Samples were further analyzed through the Personalis DNA pipeline for small variant calling (SNVs, InDels) and copy number changes. We used Qiagen Ingenuity Variant Analysis (IVA) [52] and QCI-I Translational platform [53] for variant calling. Variants filtering was done based on call quality, and allele frequency in known

populations using 1000 Genomes Project [54, 55], allele frequency community (including gnomAD and CGI), ExAC, and NHLBI ESP [56]. The de-identified FASTQ files for 4 whole exome sequencing were submitted to Stanford Digital Repository and available via the provided URL link (https://purl.stanford.edu/jd768nw9509).

### Statistical analysis

Data, including raw count, sum, and average ±SD are presented in the results section. We chose to use Fisher's exact test in the analysis of contingency tables to compare ME/CFS and healthy group, as it is more appropriate for small sample sizes in comparison to the chi-square test or G-test of independence. A probability value of P<0.05 was considered significant.

## Results

### Literature-based morphological characteristics of unstimulated and stimulated peripheral blood mononuclear cells using transmission electron microscopy

In order to better understand the cell morphologies observed in our TEM study of PBMCs isolated from ME/CFS patients, we first performed a literature-based study of TEM images of PBMCs. PBMCs encompass a heterogeneous cell population comprising monocytes (which can be differentiated to macrophages and dendritic cells) and lymphocytes (T cells, natural killer cells, and B-cells). However, most PBMC purification methods contain a considerable amount of platelet contamination [57] as well as traces of erythrocytes, and low-density granulocytes (neutrophils, basophils, and eosinophils) [58, 59]. The existence of a round nucleus in PBMCs can help easily distinguish most immune cells under TEM from erythrocytes and platelets, which have no nuclei and from granulocytes, which show a lobulated segmented nucleus [60] as well as various types of cytoplasmic granules.

However, identifying each cell type in this heterogeneous cell population using TEM can be challenging. To better identify PBMC subpopulations, we first characterized TEM images that are available in NCBI, hematology references and other online sources (Table 3) and compared our TEM images with those previously annotated [61–88]. We included a brief description for each immune cell type from the literature. Table 3 provides TEM-based morphological characteristics of unstimulated (resting) and stimulated T cells, B cells [65], natural killer cells, monocytes, macrophages, dendritic cells, and platelets. In the resting state, cells are round with a large nucleus with clear cytoplasmic organelles including the Golgi apparatus, endoplasmic reticulum, and mitochondria.

Compared to unstimulated cells, previous studies indicate stimulated PBMCs undergo morphological changes, depending on the type of stimulus and cell [92–98] (Table 3). One striking difference is the increase in the length and number of membrane protrusions such as microvilli and invadosome-like structures (invadopodia and podosomes) [99] in stimulated cells (Table 3) [100, 101]. Microvilli are involved in a variety of cellular functions such as absorption, secretion, cellular adhesion, extravasation and mechanotransduction as well as immunological synaptosomes for immune cells [102] (S3 Fig). It has also been reported that the tips of microvilli in stimulated T cells contain clusters of T cell receptors (TCR)s, enabling the cells to recognize antigenic moieties on target cells [103–105].

We also included transmission electron micrographs from two main types of cell death from literature, apoptosis and necrosis in immune cells (Table 4) [106–108]. Apoptosis is a programmed cell death, which can be triggered either intrinsically from intracellular signals generated from cellular stress or extrinsically via extracellular ligands binding to cell surface

**Table 3. TEM-based morphological characteristic of immune cells.**

| Cell Activation Status | Cell Fraction | Cell Type | Description | Reference |
|---|---|---|---|---|
| Unstimulated (Resting) | PBMCs | T cell | 5–10 μm in diameter possessing a large nucleus and a relatively thin rim of cytoplasm border that contains few mitochondria, ribosomes, and lysosome | [61, 62] |
| | | B cells | Resting B cell is very similar to T cell: 5–10 μm in diameter possessing a large nucleus and a relatively thin rim of cytoplasm border that contains few mitochondria, ribosomes, and lysosome | [62, 65, 69] |
| | | Natural killer cell (NK) | Resting cells are relatively small and round, 5–7μm in diameter and possess cytolytic granules, and contains few mitochondria and ribosomes | [63, 64, 66] |
| | | Monocytes | 12–20μm in diameter, one of the largest peripheral blood cells, with irregular kidney-shaped nucleus with thinly dispersed chromatin pattern and small amounts of rough endoplasmic reticulum and polyribosomes. Majority of monocytes are round with smooth edges, but some have pseudopod-like protrusions | [67–70] |
| | | Macrophage | 2–20μm or in diameter with randomly kidney- or spindle-shaped small nucleus with one or two distinct nucleoli and large cytoplasm, with vacuoles present at cell periphery and often features distinct lamellipodial extensions in all directions | [68, 69, 71] |
| | | Dendritic cell | 12–20μm in diameter, contain large numbers of mitochondria and exhibit long dendritic processes (pseudopodia) and have a rough cellular membrane | [72] |
| | Platelet | Platelet | 5–3 μm in diameter, with no nucleus, but contains multiple vesicles and granules | [73] |
| | Granulocytes | Basophil | 8–11μm in diameter and contain large secretory granules and lipid bodies | [74, 75] |
| | | Eosinophil | 11–14 μm in diameter, often have a bilobed nucleus and contain numerous distinctive ellipsoid granules with a linear electron dense crystalline core | [62, 76] |
| | | Neutrophil | 9–12 μm in diameter, contain small granules of various types and a lobulated nucleus | [75, 77] |
| | | Mast cell | 8–20 μm in diameter contain large numbers of cytoplasmic granules that are smaller in size than those of basophils. Mast cell surface tends to have narrow elongated protrusions | [78, 79] |
| Stimulated | PBMCs | T cell | Activated cells have rougher cell membrane with relatively little rough ER and filled with free ribosomes | [69, 80, 81] |
| | | B cell | Activated cells have abundant cytoplasm filled with an extensive rough endoplasmic reticulum (ER) | [69, 82] |
| | | Natural killer cell (NK) | Activated cells have migratory morphology exhibiting generally more elongated and irregular shapes with larger pseudopods | [66, 83, 84] |
| | | Macrophage | Full of cytoplasmic vacuoles and phagosomes containing organic (cellular, microbial) as well as inorganic foreign materials and have a leading pseudopodium in one direction | [85, 86] |
| | | Dendritic cell | Show a highly vacuolated appearance and large number of long dendritic processes to give them a very large surface-to-volume ratio for antigen presentation | [87–89] |
| | Platelet | Platelet | Dense granule membrane proteins incorporate with the platelet plasma membrane, formation of small cell protrusions, cytoskeletal proteins rearrangement leading to more ameboid shape | [73] |

Please note that macrophages and mast cells are mostly tissue-resident cells and less likely to be found in periphery [90, 91].

'death receptors' (DR) [109, 110]. This latter stimulus triggers major morphological changes in the cell such as blebbing, cell shrinkage, nuclear fragmentation, chromatin condensation and margination, cytoplasmic vacuolization, and cell lysis, originally observed by transmitted light and electron microscopy [111–114] (Table 4, and S1C Fig). Unlike apoptosis, necrosis is considered unregulated cell death, which can be caused by severe damage to the cells from internal or external stresses such as mechanistic injuries, chemical agents, or pathogens. In necrosis the cells often swell, rapidly lose membrane integrity and release cellular products into the extracellular space [108, 109, 115] (Table 4). We used the chromatin condensation and cytoplasmic shrinkage with an intact plasma membrane as the major features for annotating cells undergoing apoptotic cell death [116] (Table 4). In some cells we could also see increased vacuolization of the cytoplasm and marginalization of the condensed chromatin, micronuclei formation and apoptotic bodies [117]. Necrotic cells were identified based on the loss of membrane integrity,

**Table 4. TEM-based morphological characteristics of two major types of cell death: Apoptosis and necrosis.**

|  | Description | Reference |
|---|---|---|
| Healthy cell | Exhibits intact cell membrane and nucleus with cytoplasm containing morphologically normal mitochondria and other cell organelles | [62] |
| Apoptotic cell | Distinct chromatin condensation and marginalisation, blebbing of cytoplasm and an intact plasma membrane | [118] |
| Necrotic cell | Loss of chromatin, disrupted plasma membrane, vacuolisation and electron lucent cytoplasm | [119] |

disintegrated cell membrane, cytoplasmic swelling and vacuolization (which are absent in the apoptotic cell), low cytoplasm density and loss of chromatin (Table 4) [107].

## Increases in apoptotic and necrotic cell death in stimulated T cells of ME/CFS patients

We next investigated the impact of immune cell stimulation on both T cells as well as PBMCs lacking T cells by TEM. Morphological and ultrastructure characteristics as well as cell death were investigated in the four subjects (discordant twins and unrelated age-, gender-, and BMI-matched extremely affected and unaffected individuals).

T cells were isolated from the PBMCs via negative selection (see methods) and stimulated for 12h using anti-CD3/CD28 beads. TEM was utilized to analyze the ultrastructural images of stimulated T cells. Stimulated T cells could be distinguished from unstimulated cells based on the increase in the length and number of microvilli, interdigitating fingerlike processes, and immune synapse [94–98] (S3A and S3D–S3F Fig). Healthy viable cells were identified based on their normal morphology including intact cell and nuclear membranes (Fig 1E), whereas apoptotic cells displayed nuclear shrinkage, chromatin condensation, apoptotic bodies with intact plasma and nuclear membrane fragments (Fig 1B, 1F and 1G and S1C Fig) similar to those described in Table 4. Necrosis was marked by disintegrated cell membrane, low cytoplasm density, and swollen organelles (Fig 1C and 1H). Cell counting and morphological analysis was performed from electron microscopic images at 200x (Fig 1A–1C) or 500-1500X magnification (Fig 1E–1H). The mean number of evaluated cells per participant at 200X or 500-1500X magnification was 1247±128 and 144±28, respectively. The total number of evaluated cells per sample and the summary of apoptotic and necrotic cells are included in S1 Table.

Both apoptotic and necrotic cells were present within all the stimulated T cell samples, with up to 10.9% and 20.2% of the cells expressing apoptotic or necrotic features, respectively (S1 Table). To assess any potential differences between the ME/CSF and healthy controls, we performed Fisher's exact test by creating a contingency table, comparing the cell viability between ME/CFS and healthy control (live vs apoptotic, and live vs necrotic) (S2–S4 Tables). The number of apoptotic and necrotic cells was markedly higher in stimulated T cells collected from ME/CFS patients at both resolutions. At 200x, there was a 2- and 1.5-fold increase of apoptotic and necrotic cells (p-value = 6.86e-07, and 0.0031, respectively) and at 500-1500x we saw a 3.2- and 2.7-fold increase (p-value = 0.00058, and 0.00022, respectively) (Fig 1D and 1I and S2 Table).

We also compared morphological changes of activated T cells within each pair: the identical twins discordant with moderate ME/CFS as well as the unrelated pair. Compared to unrelated healthy control, the stimulated T cells from the extremely severe ME/CFS patient showed a marked increase in both apoptotic and necrotic cell death (at 200X: 3.7- and 2.6- fold increase and at 500-1500x: 5.8- and 3.1-fold increase, respectively) (at 200x: p-value = 7.76e-08, and 4.614e-05, at 500-1500x: p-value = 0.002. and 0.01, respectively) (S3 Table). The electron

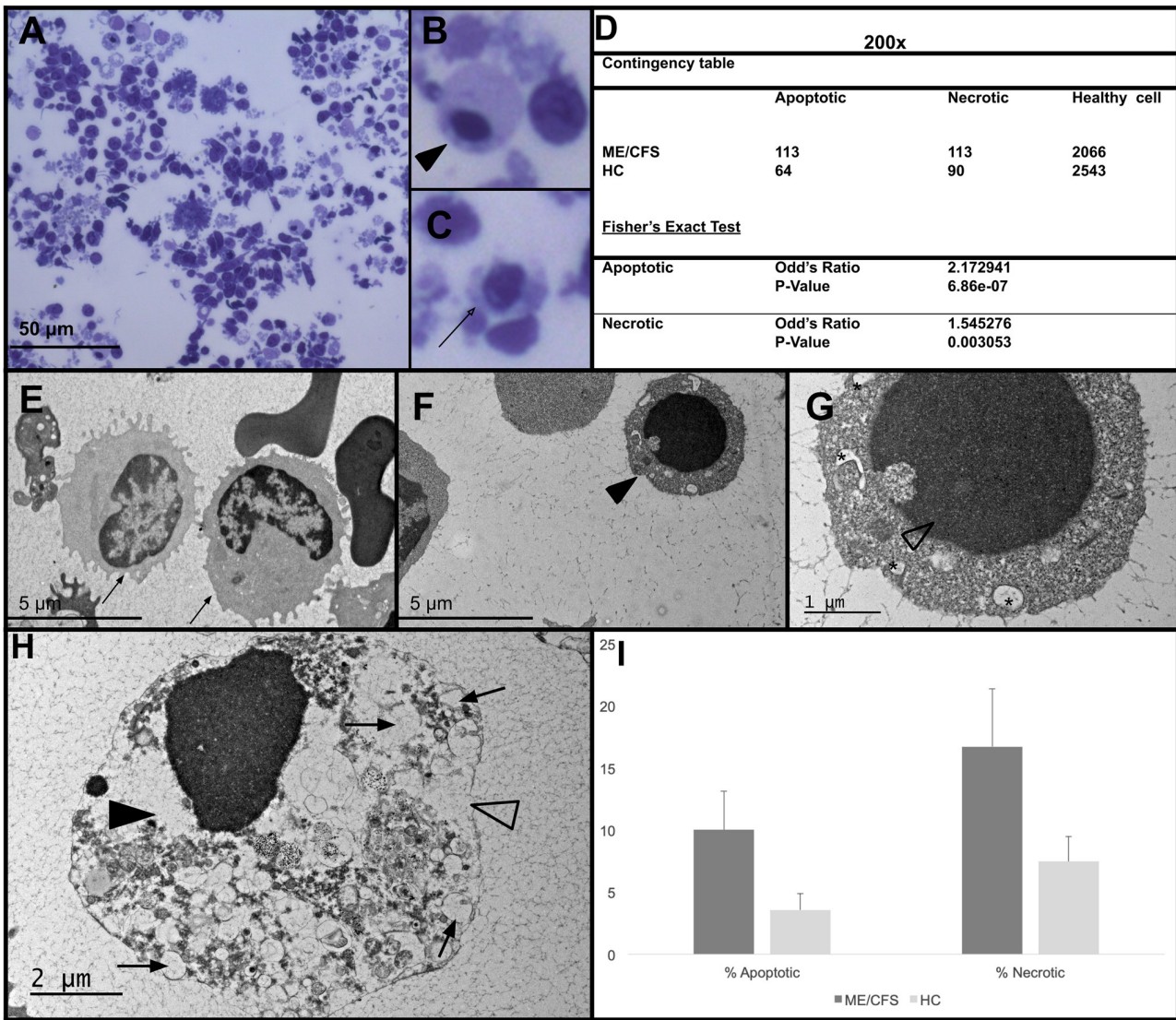

**Fig 1. TEM micrographs at 200x and 1000x magnification exhibiting apoptotic and necrotic cell death in stimulated T cells.** T cells were isolated and stimulated with anti-CD3/CD28 coated magnetic beads and then fixed 12h later and subjected to TEM imaging. (A) A representative TEM micrograph of stimulated T cells (T+Act) at 200x. (B) An apoptotic cell (Arrowhead) with distinct chromatin condensation. (C) A necrotic cell (Open arrow) with poorly defined edges and loss of plasma membrane integrity. (D) Contingency table showing the number of apoptotic, necrotic, and healthy cells present in the stimulated T cells of ME/CFS patients and healthy controls (HC) at 200x magnification. Fisher's exact test shows that there is a significant increase in total apoptotic and necrotic cells in the ME/CFS cohort (p-value ≤ 0.05). (E) A healthy T cell (arrows), with intact cell membrane and nucleus. (F, G) An apoptotic cell (arrowhead) exhibiting chromatin condensation (open arrowhead) and cytoplasmic shrinkage. (H) A necrotic cell displaying electron lucent cytoplasm (arrowhead), loss of plasma membrane integrity (open arrowhead) and increase in vacuolization (arrow). (I) Percentage of apoptotic and necrotic cells in ME/CFS subjects and healthy controls. TEM micrographs were scored, apoptotic and necrotic cells were counted and expressed as a mean percentage of cells. Magnification was between 500x and 1500x.

micrographs of stimulated T cells from the moderately affected twin showed a significant but less severe increase in necrosis at higher magnification (p-value = 0.0022) (S3 Table). Although the sample size is small, these results suggest severe ME/CFS is associated with significant increases in both apoptosis and necrosis whereas moderate forms may preferentially affect necrosis.

We did not systematically investigate autophagic cell death (ACD) in our samples, which is distinct from apoptotic cell death due to the absence of chromatin condensation and

accumulation of large-scale autophagic vacuoles in the cytoplasm [120]. However, we could detect autophagic cells, displaying autophagic-like vacuoles filled with amorphous materials, and membranous inclusion (S1D Fig).

## Apoptotic and necrotic cell death in the unstimulated and stimulated PBMC subpopulation lacking T cells

The unstimulated and stimulated PBMC subpopulation lacking T cells were also examined using TEM. The unstimulated PBMCs were studied in identical twins and the unrelated healthy control, whereas the stimulated cells were only studied in the unrelated extremely severe ME/CFS case and healthy control. Stimulation was performed by incubating the cells with 100nM PMA for 12 hours and apoptotic and necrotic cells were identified using the criteria mentioned above (Table 4). Cell counting and morphological analysis was performed using TEM micrographs at magnification of 200-1500X. The total number of evaluated cells per study group and the summary of apoptotic and necrotic cells are included in S4 Table.

For unstimulated PBMCs lacking T cells, comparison of the apoptotic and necrotic cell death between the identical twins discordant for ME/CFS revealed no significant differences in apoptosis (p-value = 0.1, Fisher's exact test), and only a slight increase in necrosis (p-value = 0.06) (S4 Table). Stimulated PBMC lacking T cells from the unrelated pair also did not show any marked difference in apoptotic cell death ratio. However, there was slightly higher necrotic cell death in PBMCs lacking T cells in the extremely severe ME/CFS compared to the unrelated healthy control (p-value = 0.065) (S4 Table).

## Mitochondrial structural characteristics in unstimulated and stimulated PBMC subpopulations isolated from ME/CFS and healthy control

TEM is a standard imaging method to directly observe the ultrastructure of subcellular organelles, such as the nucleus, smooth endoplasmic reticulum (SER), rough endoplasmic reticulum (RER), Golgi apparatus, various endosomes, lysosomes, ribosomes, mitochondria, and peroxisomes as well the spatial relationship between organelles.

TEM has revealed that normal mitochondria are 0.75–3 micrometers in length, containing an inner and outer membrane with distinct function [121, 122]. The number of mitochondria varies based on cell type, context and activation status, with cells that require high energy demand generally having greater numbers of mitochondria [29, 123]. TEM imaging also has shown that mitochondria can adopt a wide range of shapes, as they are constantly dividing (fission) and fusing [34]. At very high TEM magnification, five mitochondrial morphologies can be examined including normal, normal–vesicular, vesicular, vesicular–swollen: and swollen (Table 5) [124–126]. Vesicular mitochondria arise due to a structural transformation of the inner membrane of a normal mitochondrial into multiple vesicular matrix compartments, which will further lead to the release of proteins such as cytochrome c from the intermembrane and intracristal regions (Table 5). This transformation initiates cellular apoptosis, which eventually leads to the loss of the mitochondrial membrane potential ($\Delta\Psi$m) and swollen mitochondria [126] (Table 5).

We first investigated mitochondrial numbers and morphology in the stimulated T cell population collected from the discordant twins as well as the extremely severe ME/CFS subject and healthy control (Fig 2, S5 Table). To conduct a semi-quantification assessment of mitochondrial number, size, and their interior ultrastructure, we only used magnification between 300-2500x, which would reveal all those parameters within an intact cell (Fig 2). At this resolution, we could easily identify normal mitochondria from those exhibiting the vesicular/compartmentalized or swollen ultrastructure in TEM micrographs (Fig 2, Table 5). However,

**Table 5. TEM-based morphological characteristics of mitochondria with normal or abnormal ultrastructural appearance.**

| | | Description | Reference |
|---|---|---|---|
| Normal | Normal | Exhibits an intact outer membrane and matrix with an inner boundary membrane connected to lamellar cristae via crista junctions | [125, 126] |
| Vesicular/ Compartmentalized | Normal-vesicular | While mostly exhibits a healthy mitochondrial morphology, in small area the connection between inner boundary membrane and lamellar cristae at crista junctions changes, forming separate vesicular matrix compartments, which disrupt the intact structure of matrix and its function | [125, 126] |
| | Compartmentalized (Compartmentalized/vesicular) | Exhibit larger number of separate vesicular matrix compartments and circular or rounded cristae throughout the mitochondrial body | [125, 126] |
| Swollen | Vesicular-swollen mitochondria. | Substantial formation of vesicular matrix compartments, leading to matrix fragmentation. Vesicular-swollen mitochondria occurs during the release of cytochrome c and leads to apoptosis | [125, 126] |
| | Swollen | Displays expanded matrix space, translucent matrix, and fragmented or disorganized crista | [125, 126] |

characterizing normal–vesicular morphologies could not be performed at this magnification. Vesicular/compartmentalized mitochondria display vesicular matrix compartments in one domain of a mitochondrion, whereas another domain looks normal (Fig 2B, Table 5). Swollen mitochondria exhibited fewer cristae, expanded matrix space, and less dense staining of the matrix (Fig 2C, Table 5).

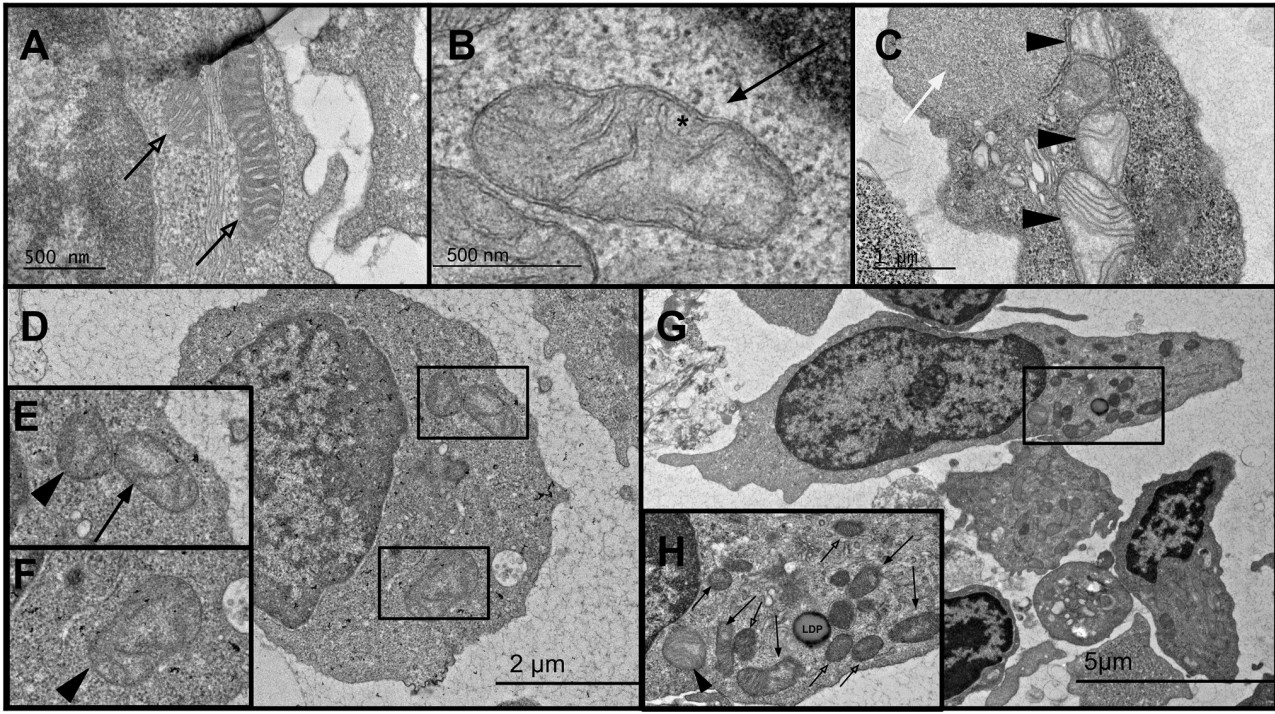

**Fig 2. TEM analysis of the stimulated T cells to identify changes in mitochondrial morphology upon stimulation.** A) A representative TEM image of normal mitochondria (open arrow), with dense staining of the inner matrix with an intact outer membrane at 10000x magnification. B) TEM image of a vesicular mitochondria (arrow) exhibiting an inner membrane enclosing separate vesicular matrix compartments (*) and rounded cristae at 10000x magnification. C) Displaying 3 swollen mitochondria (arrowhead), which show expanded matrix space and disorganized crista at 5000x magnification. Note a giant intracellular lipid droplet-like vesicle (white arrow) at the top-right corner. D) Representative image of several abnormal mitochondria within a single PBMC cell at 2000x magnification, E) zoomed in image of two abnormal MT within a cell, one vesicular and one swollen, F) zoomed in image of a swollen mitochondria. G) Representative image of the number of mitochondria found within a PBMC at single cell resolution at 1200x magnification, H) healthy, vesicular, and swollen mitochondria are all present within one single cell, as well as a lipid droplet-like vesicle (LDP).

The total number of evaluated cells and mitochondria (with normal, vesicular/compartmentalized, or swollen ultrastructure) per individual are summarized in S5 Table. Mitochondria with vesicular/ compartmentalized and swollen morphologies were considered abnormal. We ran Fisher's exact test by creating a contingency table, comparing the association between ME/CFS and the healthy control and mitochondria morphology (normal, vesicular/compartmentalized or swollen) (S6 Table). The average number of mitochondria per cell for each group can be found in S5 Table. On average, ME/CFS cells contained 9.1 and healthy control cells had 8.3 mitochondria per cell, a result which is not statistically significantly different and consistent with previous findings [25, 38]. However, ME/CFS stimulated T cells showed substantially higher level of swollen and abnormal mitochondria (vesicular/ compartmentalized and/or swollen mitochondria) (1.9- and 1.8-fold increase with a p-value = 0.00004 and p-value = 0.003, respectively) (S6 Table). Within each pair, both the ME/CFS twin and the extremely severe ME/CFS patient showed remarkably higher numbers of swollen mitochondria (2 and 2.1-fold, respectively) (p-value = 0.0003, and 0.011, respectively) (S6 Table).

To assess the extent of mitochondrial morphological abnormalities within each cell, we scored the number of cells that contain 3 or more swollen mitochondria (S7 Table). Interestingly, the number of stimulated T cells carrying more than 3 swollen mitochondria per cell was significantly higher in the ME/CFS group, (25% of the ME/CFS vs 5.3% healthy control cells contained more than 3 swollen mitochondria per cell) (p-value = 0.001) (S8 Table). We also noticed a significant increase in individual stimulated T cells that carried 6 or more morphologically abnormal mitochondria (showing vesicular/compartmentalized and/or swollen pattern) (27.5% of the ME/CFS versus 8.5% healthy control cells, p-value = 0.006) (S8 Table). Interestingly, when compared to moderately affected twins, the percentage of stimulated T cells from the extremely severe ME/CFS carrying more than 6 abnormal mitochondria per cell was 2-fold higher (18% vs 38%, respectively) (S7 Table). These results suggest a positive correlation between disease severity and the extent of mitochondrial damage at single cell level after stimulation (S7 Table).

TEM high resolution enabled us to identify other subcellular organelles such as Golgi apparatus with visible cisternae (S4A Fig), endoplasmic reticulum (S4B Fig), lysosome-like vesicles with electron-dense cores indicative of high protein concentration (S4C Fig), and centrioles and their multiple microtubule (MT) triplets (S5C Fig), surrounded by electron dense pericentriolar material (S5D Fig). We could also identify different vesicles present in the PBMC subpopulation such as autophagosomes (S4D Fig), multilamellar bodies (MLBs) [127, 128] (S5A Fig), multivesicular bodies (MVBs) [129–131] (S5B Fig) and vesicles containing electron dense material, resembling autophagosome (S4D Fig). Additionally, we observed direct and indirect interaction between platelets and PBMCs (S3B and S3C Fig), and the formation of immunological synapse (S3 Fig).

## Elevated intracellular giant lipid droplet-like organelles in the extremely ill ME/CFS stimulated PBMCs subpopulation

Lipid droplets, oil bodies, or adiposomes are highly dynamic intracellular organelles with biological roles significantly broader than neutral lipid storage within a cell. Lipid droplets are found in all eukaryotic organisms and involved in intra- and extra-cellular fatty acid trafficking, cellular metabolism, energy homeostasis, assembly platforms for protein binding and degradation, chromatin remodeling, gene expression, and other biological signaling pathways, such as endocannabinoids synthesis [132], and their dysfunction has been linked to many diseases. Abnormality in lipids, such as sphingolipids and phospholipids, has been previously observed in ME/CFS patients [28, 133, 134].

The size of lipid droplets can vary from 20nm to 100 µm [135]. In our study, we observed lipid droplet-like organelles in the PBMC population in the TEM micrograph (Fig 3 and S2 Fig). The stimulated PBMC lacking T cells subpopulation (P-T+Act) displayed the highest number of intra- and extracellular lipid droplet-like organelles (S9 Table). TEM micrographs

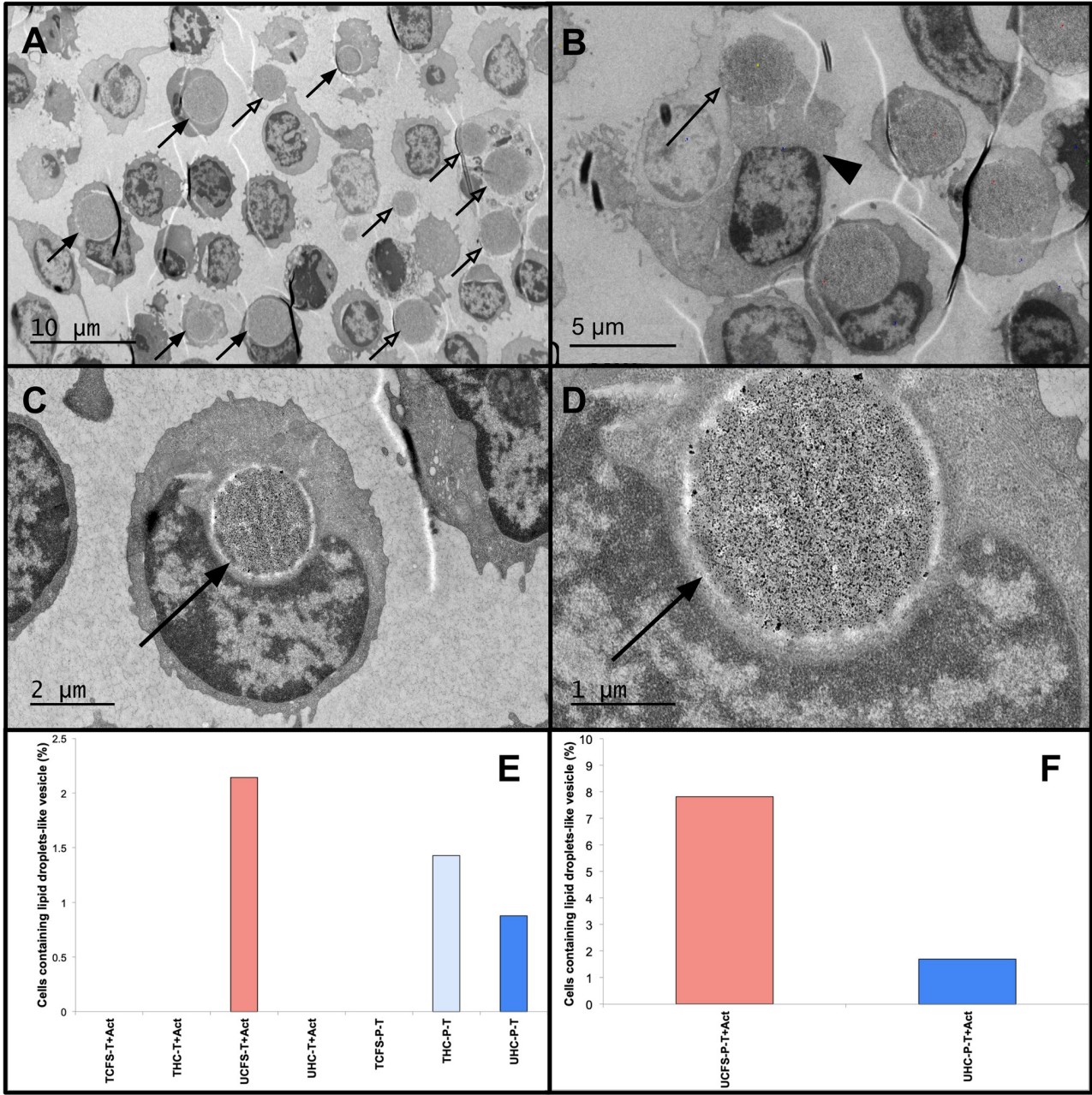

**Fig 3. TEM analysis of giant lipid droplet-like vesicles within PBMC subpopulations.** The micrographs from stimulated PBMC lacking T cells from the unrelated extremely severe ME/CFS patient (UCFS-P-T+Act) showing: (A) the presence of intracellular giant lipid droplet-like vesicles (arrows) and extracellular giant lipid droplet-like vesicles (open arrows) at 500x magnification, (B) an immune cell releasing/engulfing (arrowhead) a giant lipid droplet-like vesicle (open arrow) at 1000x magnification, (C) image of an immune cell containing a giant lipid droplet-like vesicle (arrow) at 2000x magnification, (D) the magnified image of the giant lipid droplet-like vesicle shown in C. (E) Percentage of stimulated T cells and unstimulated PBMC lacking T cell subpopulation that contain intracellular giant lipid droplet-like vesicles. (F) Percentage of stimulated PBMC lacking T cell subpopulation that contain intracellular giant lipid droplet-like vesicles in extremely severe unrelated ME/CFS and unrelated age- and gender-match healthy control.

showed a single very large 2–5 μm intracellular lipid droplet-like organelles within a subset of these cells, which comprised the majority of the cell volume (Fig 3A–3D and S2B, S2D and S2F Fig), notably this is similar to those observed in adipocytes [136]. The percentage of cells carrying these giant lipid droplet-like vesicles was remarkably higher in the extremely severe ME/CFS patient (7.8% in severely ill ME/CFS vs 1.6% of healthy control (fisher exact test P-Value = 0.02) (Fig 3A, 3E and 3F) (S10 Table).

The percentage of lipid droplet-like organelles appears to be cell type-specific and positively correlates with ME/CFS disease severity (Fig 3, S2 Fig, S9 and S10 Tables). Of 459 counted stimulated T cells in the entire cohort, we only observed lipid droplet-like vesicles in 3 cells of the extremely severe ME/CFS patient (which account for 0.6% of the total cell count) (Fig 3E, S9 Table). The size of the lipid droplet-like organelles was also considerably higher in number and smaller in size (100nM- 2uM) in those 3 stimulated T cells, mostly resembling those found in foam cells (S2A and S2C Fig). Few cells in the unstimulated PBMC lacking T cells contained lipid droplet-like organelles, indicating that the presence of lipid droplets may be associated with the immune cell activation and a specific class of immune cells (Fig 3, S2 Fig and S9 Table).

## Platelet ultrastructure analyses using TEM

Platelets are small, short-lived (7–10 days), anucleated blood cells, and an indispensable part of our immune system and many vital biological functions such as the prevention of bleeding and the maintenance of vascular integrity and hemostasis [137] and growth and development [138]. Dysregulation in platelets has been associated with many pathological conditions such as inflammatory, autoimmune, and cancer diseases as well as impairment in growth and development, angiogenesis, and wound healing [139].

Platelets are 2–3 μm in diameter and are readily identified using TEM because they lack a nucleus (Table 3). Platelets also contain open canalicular system (OCS), an elaborate invaginated internal membrane structure, which is made of tunneling network of surface-connected channels and serves as a pathway to transport granular substances to the extra-platelet environment [140] (Fig 4A). Under TEM, we could observe other intracellular organelles within platelets, such as mitochondria (which provide the energy needed for platelets via aerobic respiration), a dense tubular system (DTS) [141], Golgi apparatus [142], and glycogen granules, which could serve as a reservoir for energy production at the early stage of platelet activation (Fig 4A and 4B) [141].

Platelets were highly abundant in the PBMC population, especially in the PBMC lacking T cells (Fig 4D). Using TEM, we could also observe very large platelets, which we refer to as giant platelets (Fig 4C), as well as aggregates of many platelets (Fig 4D and 4E, S11 Table) [143], a few of which were much larger than a normal platelet clump, forming a giant rosette-like structure 8–12 microns in diameter (Fig 4D–4I). At 1200X resolution, it seems that these rosette-like structures are mainly formed due to the adhesion of activated platelets to each other and to the fragmented dead platelets or other cellular debris (Fig 4G and 4I). At the periphery of those rosette structures, platelet-derived microparticles (PMPs: small fragments released from platelet cell membranes because of cell activation or apoptosis)-like structures are also visible (Fig 4H).

In our micrographs, giant rosette-like platelet aggregates are only present in the stimulated T cells (Fig 4) and not in unstimulated and stimulated PBMC lacking T cells (S11 Table), indicating they are predominantly associated with T cell activation. The number of large platelets, platelet clumps (which were counted when more than 5 small platelets adhered to each other), and giant rosette-like platelet aggregate structures can be found in S11 Table. We saw an

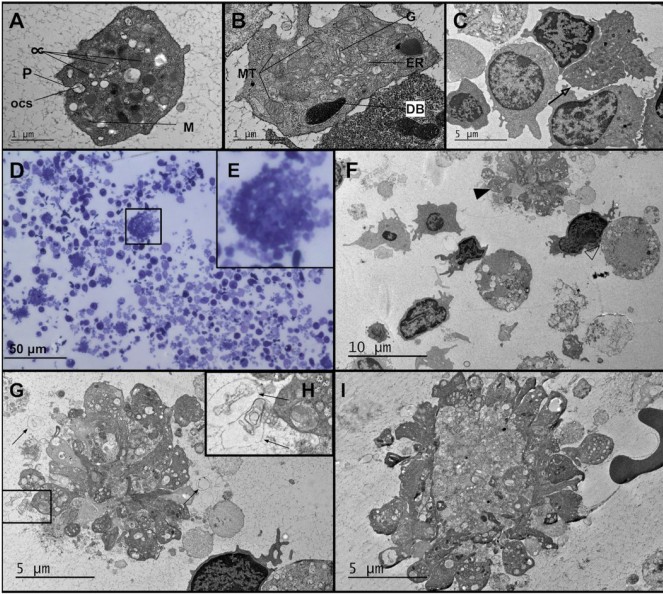

**Fig 4. TEM micrographs of platelets, giant platelets and platelet aggregates present in PBMC subpopulation.** (A) Equatorial section of a platelet, featuring α-granules (α), microfilaments (M), mitochondria (MT) and pores (P) of the open canalicular system (ocs). (B) Equatorial section of a platelet featuring dense bodies (DB), mitochondria (MT), endoplasmic reticulum (ER) and a Golgi body (G). (C) A giant platelet 8 μm in diameter (open arrow), surrounded by a few PBMCs. (D) A representative TEM image of stimulated T cells (T+Act), and several small and giant platelet aggregates at 200x. The micrograph displays the size of a giant platelet aggregate compared to the rest of PBMCs and platelet clumps. (E) The magnified image of the giant platelet aggregate shown in D. F) TEM image depicting a giant platelet aggregate on the top corner (arrowhead), and an immune synapse between a healthy cell and a necrotic cell (open arrowhead). G) Zoom-in image of F, showing a giant platelet aggregate, possibly formed after being activated by thrombin or other factors, H) Platelet-derived microparticles (PMP) (arrow). PMP like structures can be formed upon platelet activation. I) A giant platelet aggregate that has formed a rosette like structure around a central core.

increase in platelet clumps and giant rosette-like platelet aggregates in stimulated T cells from ME/CFS patients, however, the data was not significant, possibly due to small sample size (S12 Table). We also observed an increase in large platelet and platelet clumps in stimulated T cells from the extremely severe ME/CFS patient (2.2- and 2.1-fold), compared to unrelated healthy control (p value = 0.0164 and 0.17, respectively) (S13 Table), which may indicate platelet hyperactivation could be associated with illness severity. Genetics and environmental factors such as medication could also affect platelet function and therefore lead to platelet hyperactivation. Of interest, we also detected a monocyte-like cell engulfing a platelet, a phenomenon which has been described by others as one means for monocyte differentiation into pro-inflammatory M1 proinflammatory macrophage [144] (S1A Fig).

## Whole-exome sequencing results

We performed whole exome sequencing using Personalis ACE Clinical Exome sequencing platform to a depth of ~70X and Qiagen QCI-I Translational for data interpretation and variant calling. These results revealed a rare homozygous SMPD1 (sphingomyelin phosphodiesterase 1) variant with uncertain significance (c.808G>A; p. Gly270Ser, CAD score: 23.7, ExAC Frequency: % 0.023, ClinVar Accession: RCV000382375.1) in the extremely severely ill ME/CFS patient. The observed variant resides in the metallophosphatase (MPP) domain of the SMPD1 gene. Mutations in SMPD1 are associated with sphingomyelin lipidosis, also called sphingomyelinase deficiency or Niemann-Pick disease (NPD) type A/B, a rare lipid storage

disorder with autosomal recessive inheritance. NPD type A/B can cause a spectrum of disease with variation in severity and symptomology even among members of the same family [145]. We did not find any rare damaging homozygous SMPD1 variant in the identical twins or unrelated healthy control in their whole exome seq data. These results raise the possibility that the variants in the SMPD1 gene may be responsible for the lipid droplet increase observed in the extremely severe ME/CFS patient.

## Discussion

Immune system dysregulation has been accepted as one of the pathological bases of ME/CFS, therefore, blood collected from ME/CFS patients may be valuable to study this multisystemic debilitating disorder. The aim of this study was to examine morphological changes in peripheral blood mononuclear cells and alterations of mitochondria, other cellular organelles, and ultra-structures in ME/CFS patients using transmission electron microscopy. Currently, there are a very limited number of papers focusing on TEM imaging of cellular morphology, ultra-structural characteristics, and number of mitochondria in the context of ME/CFS at single-cell resolution [39, 146], only one was focused on PBMCs [40].

We first investigated the levels of apoptotic and necrotic cells present in the unstimulated and stimulated PMBCs. We found significantly higher levels of both necrotic and apoptotic cell death in stimulated T cells from ME/CFS patients, which was positively correlated with ME/CFS disease severity. Even though our sample size is very small, our results are consistent with previous work on ME/CFS, suggesting broad T lymphocyte dysfunction and immune dysregulation may play a major role in ME/CFS pathogenesis [5, 21, 26, 94, 147]. A previous study has also shown that compared to healthy controls, ME/CFS patients have higher levels of apoptotic neutrophils [148]. Our results suggest increased apoptosis, as well as necrosis extend to other cell types.

Accelerated exacerbated cell death in antigen-stimulated T cells in ME/CFS patients could lead to chronic persistent infection and reduced capacity to fight against invading pathogens. The existence of chronic microbial infections in ME/CFS has long been proposed as one of the primary causes of disease pathophysiology [9, 149].

The increase in apoptotic and necrotic cell death in stimulated T cells of ME/CFS patients was further confirmed with our results on mitochondrial morphological features. While the average number of mitochondria per cell was not statistically different between groups, we saw remarkably elevated levels of swollen and morphologically abnormal mitochondria in ME/CFS stimulated T cells. Of interest, the extent of the mitochondrial damage, which was defined based on the total number of affected mitochondria per sample and per single cell, was positively correlated with ME/CFS disease severity. The higher number of swollen or morphologically abnormal mitochondria per cell might exponentially exacerbate the impairment in energy production and metabolism. Mitochondrial vacuolation, compartmentalization, swelling as well as other structural abnormalities can impair mitochondrial function and membrane integrity, which lead to the release of cytochrome C, apoptosis and in severe cases to necrosis. Conversely, internal, or external apoptotic signals can be propagated to the mitochondria, leading to mitochondrial dysfunction and cell death [150]. Our results on mitochondrial morphological characteristics of immune cells are mostly consistent with the limited number of studies performed on ME/CFS patients, including those of Behan et al. also reported a substantial increase in mitochondrial abnormalities in muscle cells from ME/CFS patinets [38]. It is also in line with the Fisher et al observation of no change in mitochondrial number, but a greater mortality rate for ME/CFS EBV-mediated immortalized lymphocytes due to a complex V impairment [25]. Additionally, using fluorescence-based confocal imaging approaches,

Mandarano et al did not observe differences in mitochondrial mass and membrane potential in unstimulated and stimulated CD4+ T cells from ME/CFS patients and controls [26]. The same study also showed reduced mitochondrial membrane potential and impaired metabolism in ME/CFS CD8+ T cells [26]. Plioplys et al, did not observe ultrastructural mitochondrial abnormality in ME/CFS patients—this inconsistency may be due to the differences in sample types, as they examined muscle cells, and we analyzed stimulated T cells. Despite finding no significant ultrastructural abnormalities, the authors agreed on a "presence of a possible functional mitochondrial abnormality" [39]. Lawson et al reported more condensed mitochondrial cristae membranes in ME/CFS PBMC cells. However, they did not see marked differences in crista length in patients. In our study we have examined mitochondrial number, morphology and remodeling of the inner mitochondrial membrane (such as vesicular/compartmentalized or swollen) [126] and did not study mitochondrial crista's length. Nonetheless, even though both groups have used PBMCs, we think our study design differs significantly from the 2016 PBMC TEM study. Lawson et al used previously frozen unfractionated total PBMCs, which were incubated for 72 hours for their TEM analyses. We, however, used freshly isolated PBMC cells, which were subjected to fractionation into T cells and PBMC lacking T cells subpopulation, and 12h- stimulation for downstream TEM imaging [40]. It has been shown that the differences between biospecimen, cell fraction and stimulation can impact mitochondrial structure [12, 151–154].

Accumulating data suggests that immune cell exposure to persistent antigen and/or inflammatory signals due to chronic infection, inflammation, autoimmune conditions, active viral infection, poor systemic blood circulation, and/or hypoxia can result in mitochondrial damage, an exhausted-cell state and, in severe situations, clonal deletion of the exhausted cells [6, 155–159]. Interestingly, persistent infection, chronic inflammation, autoimmune condition, poor systemic blood circulation and hypoxia are all proposed to contribute to ME/CFS etiopathogenesis [9, 149, 160–166], suggesting T cell exhaustion occurs in ME/CFS [5, 26, 167]. In physiological conditions, stimulated T cells undergo clonal expansion to differentiate into effector T cells. After eliminating the initial threat, most of the activated T cells (except the memory T cell pool) must undergo programmed cell death to resolve the inflammation and ensure innate and adaptive immune homeostasis [168]. Our TEM results on exacerbated cell death in CD3/CD28-stimulated T cells from ME/CFS patients suggests an association between exhausted T cells and a reduced capacity to fight against invading pathogens and resolve the inflammation [26, 155, 169].

Although we only found a borderline increase in necrotic cell death in PBMC lacking T cells subpopulation from ME/CFS patients, longer incubation times and larger sample sizes might provide more substantive results. The increase in T cell death upon antibody stimulation suggests that the T cell subpopulation in ME/CFS might be more susceptible to activation-induced cell death (AICD)–a consequence of repeated stimulation through the CD3/TCR (T cell receptor) signaling [170]. Further studies combined with advanced apoptosis assays will help evaluate these findings, investigate the cellular events associated with the T cell death, and ferret out the most susceptible subsets, as well as determine if a potential causal relationship exists between T cell exhaustion and ME/CFS pathophysiology. A larger cohort study will also help identify ME/CFS subsets, whose stimulated T cells show delay or failure in apoptosis, leading to lymphoproliferative and autoimmune diseases. Further studies confirming and dissecting the underlying genetics and epigenetic mechanisms governing proliferation and death in stimulated T cells in ME/CFS might lead to potential biomarkers and the development of therapeutics by curtailing unwanted T cell responses.

Of interest, Behan et al also reported a mild to moderate excess of lipid, in the form of lipid droplets in the muscle cells of a subset of ME/CFS patients under TEM [38]. We also observed

a marked elevation in both intra- and extracellular giant lipid droplet-like organelles in about 5% of stimulated PBMCs lacking T cells population in the extremely severe ME/CFS patient. Plasma lipid metabolites, sphingolipids, and phospholipid abnormalities have been shown in ME/CFS and are proposed to be part of the disease's etiology [28, 133, 171, 172]. Moreover, accumulating data suggest sex-specific lipid dysregulation patterns in ME/CFS, such as higher levels of total hexosylceramides (HexCer), monounsaturated phosphatidylethanolamine (PE), phosphatidylinositol (PI), and saturated triglycerides (TG) in males, and marked reduction in total PE, omega-6 arachidonic acid-containing PE, and total HexCer in women [173]. This is consistent with our findings that the extremely severe ME/CFS patient showed a significant elevation in intracellular and extracellular lipid droplet-like vesicles. Our patients' clinical laboratory testing data also show increases in low density lipoprotein, decrease in HDL and elevated cholesterol/HDL ratio. Interestingly, in a recent paper on 20 severely ill male and female ME/CFS patients, analyses of laboratory test results also showed significantly higher level of cholesterol/HDL ratio [174] in patients.

The Naviaux et al 2016 metabolic study also found a significant abnormality in lipid molecules such sphingolipid, phospholipid, and cholesterol as well as mitochondrial metabolism in both male and female ME/CFS patients. The authors also identified sex- specific metabolic abnormalities such as those involved in methionine cycle, very long chain FAO in male and fatty acid oxidation, bile acids, vitamin B12, vitamin C/collagen in female [28]. Despite some inconsistency, lipid abnormality has been correlated to ME/CFS and our finding further supports these findings [151–154]. However, intracellular, and extracellular lipid accumulation in immune cells and its association with ME/CFS severity remains unexplored. While we could not distinguish the exact type of the stimulated PBMC cells that contain the giant lipid droplets, we believe they may be monocyte-driven macrophages. Classical monocytes contribute 5–15% to the total PBMCs pool, and monocyte-derived macrophage make up to 5% of the PBMC population [175].

Deregulation of lipid metabolism and excess intracellular lipid content, which associated with a wide range of conditions [176] (chronic bacterial, viral, and fungal infections as well metabolic, autoimmune and cancer disorders [177]) can turn macrophages into foam cells. A comprehensive list of foam cells related disorders can be found in Guerrini et al [178]. Lipid droplets are also critical for the replication of positive-stranded RNA enteroviruses [179]. Some ME/CFS outbreaks have been associated with enterovirus infection [149]. Of interest, clinical data from our extremely severe ME/CFS patient showed remarkable elevation in IgG autoantibodies against the 65 kD isoform of glutamic acid decarboxylase (GAD65), which can be due to molecular mimicry between p2C of coxsackie B-like enteroviruses and GAD65 [180]. We also identified a rare damaging homozygous SMPD1 variant in the extremely severely ill ME/CFS patient, targeting the metallophosphatase (MPP) domain of the gene. SMPD1 is responsible for the conversion of the sphingomyelin to ceramide as well as immune system regulation, apoptosis, and death-inducing signaling pathways. Mutation in SMPD1 is associated with sphingomyelin lipidosis, sphingomyelinase deficiency or Niemann-Pick disease (NPD) type A/B. Accumulations of large, lipid-laden foam cells have been reported in a wide range of cell types of NPD disease type A/B (such as liver, spleen, lymph nodes, and adrenal cortex) and are used as a diagnostic marker [181]. While our extremely severe ME/CFS patient did not have a diagnosis of NPD, our result suggests that a primary or secondary lipid storage disorder could contribute to ME/CFS pathogenesis, disease severity or comorbidity. Future studies are warranted to investigate the connection between impaired lipid storage in immune cells (and other organs) and ME/CFS pathogenesis, severity and comorbidity. Further investigation of the role of lipid droplet accumulation in the etiology of ME/CFS might help

with patient stratification and development of novel diagnosis and therapeutics focused on restoring lipid homeostasis and improving lipid metabolism.

Finally, we studied platelets ultrastructural characteristics, as they are essential components of the immune system, coagulation and vascular integrity. Our preliminary results show a significant elevation in the number of giant platelets and slight increase in platelet clumps in the ME/CFS cohort, which might be indicative of platelet hyperactivation. A few studies point to a hypercoagulation state in ME/CFS patients, which could be due to platelet hyperactivation in response to immunological disturbances or infections [182, 183]. Platelet hyperactivation may contribute to some of the major symptoms and comorbidities such as multiple chemical sensitivity (MCS) in ME/CFS patients [184], which has mostly been linked to mast cell activation, a known driver of allergic reaction. Of note, activation and degranulation of both mast cells and platelets can lead into the release of histamine, serotonin, and many inflammatory mediators, leading to a broad range of allergic type reactions (foods, aeroallergens, pharmaceuticals, and other xenobiotics hypersensitivities) and clinical manifestation (such as asthma, urticaria, rhinitis, or gastrointestinal problems) [185]. The importance of platelet activation and degranulation is consistent with the observation that serum tryptase is mostly normal in ME/CFS patients, therefore, in this group of patients platelet hyperactivation may be a contributing factor to the overreaction of their immune system to antigen or allergen [186–188]. The release of serotonin and immunomodulators from platelet-dense granules could also regulate systemic and local blood pressure and contribute to vascular endothelial dysfunction and poor blood circulation in ME/CFS patients [189–193]. Platelet-neutrophils rosettes [194], and neutrophil-erythrocyte rosette (NER) [195] structures have been reported by many research groups, however, our observation of platelet-platelet rosette structure is a new finding, which future studies can help to validate and explore their biological significance. In our TEM study, we also observed multiple ways of crosstalk between PBMCs/platelet (e.g. release of intracellular cargo or direct contact between cytoplasmic membranes of immune cells and platelets, forming immunological synapse. Further TEM studies on a larger cohort are warranted to fully investigate platelet ultrastructure, degranulation status, PMPs, PBMCs/platelet interactions and platelet-platelet rosette structure in relationships to ME/CFS pathogenesis.

We identified subcellular organelles such as Golgi apparatus, endoplasmic reticulum, lysosome-like vesicles and other vesicles including multilamellar bodies. MLBs can be found in a wide range of cell types and are mostly involved in lipid storage and secretion [127, 128]. Dysfunction in multilamellar bodies has been associated with a severe form of ichthyosis [196] and winter eczema [197]. Interestingly, dry skin, itching (pruritus), and rashes are some of the most reported common dermatologic manifestations of ME/CFS [198], therefore, it is worth investigating MBLs function in ME/CFS pathophysiology.

We also detected multivesicular bodies (MVBs). As part of endocytic machinery, MVBs are involved in sorting and separating misfolded non-wanted proteins from those that can be recycled for future use or transported to cell surface to get released to ECM as exosome. The intra-luminal vesicles (ILVs) within MVBs serve as the precursors of exosomes [129]. The role of exosomes in ME/CFS etiology has just begun to be explored [146, 199–201]. Natelson and colleagues reported that exosome-associated mitochondrial DNA is elevated in ME/CFS, and the purified exosome from patients serum promotes IL-1β secretion from microglia in cell culture model [202]. MVBs play a pivotal role in a wide range of biological functions [130], including proper cell signaling and communication with the extracellular environment, nutrient uptake and homeostasis [130, 131], reticulocyte differentiation into erythrocytes, antigen detection on mature dendritic cells [203], and transferrin receptor (TfR) secretion at the cell surface [204]. Many ME/CFS patients suffer from low levels of ferritin and chronic anemia. Serum transferrin receptor (sTfR) levels and the ratio of sTfR/serum ferritin has been proposed as a

differential diagnostic marker between anemia caused by chronic iron deficiency vs chronic inflammation [205].

MVBs also prompt the efflux of certain viruses and toxins [130]. Dysregulation in toxins efflux may impair xenobiotic metabolism, which potentially may be associated with ME/CFS pathogenesis [206] via targeting the xenobiotic receptors CAR (constitutive active/androstane receptor) [207], PXR (Pregnane X receptor) [208], LXR (The liver X receptor), FXR (The farnesoid X receptor) [209, 210], VDR (Vitamin D receptor) and AHR (aryl hydrocarbon) cascades [211–213]. This further emphasizes the value of TEM research in investigating MVBs number, morphological characteristics, and function in a larger cohort of ME/CFS patients [130] and whether defects in MVBs and exosomes contribute to this multisystemic illness.

We also identify centrioles in the PBMCs. Centrioles and centrosomes play vital roles in regulating the innate and adaptive immune response [214–216]. No studies have yet assessed the role of centrioles in ME/CFS pathology.

## Conclusion

In summary, only a handful of studies have been performed on the ultrastructural characteristics of ME/CFS muscle cells and no data are available on other cell types. Our study analyzes immune cells from ME/CFS patients for the first time and provides insights into disruption into immune cell structure and function. Although our sample size is small this study suggests new directions for characterization of morphological and ultrastructural dysregulation of affected tissues at single cell level. Our finding that the proportion of apoptosis and necrosis increase in stimulated T cells in patients with ME/CFS and that the rate of mitochondrial swelling correlates with disease severity is robust and supports previous research but needs well-adjusted replication. Elevated lipid droplet and platelet hyperactivation in the extremely severely ill ME/CFS patient highlights the roles genetics and epigenetics risk factors interplay in the onset, severity, prognosis, and comorbidity. It further reveals the power of genetics testing when combined with proper functional, diagnostic and research testing in patients with chronic complex conditions. Replicating this study with larger cohorts, more measurement time points, and perhaps a combination of other cell death assays would expand our knowledge of morphological characteristics of the immune cell in ME/CFS etiopathogenesis.

## Supporting information

**S1 Table. Quantitative analysis of transmission electron microscopy data on cellular apoptosis and necrosis in stimulated T cells.** Isolated T cells were stimulated with anti-CD3/CD28 beads for 12 h and number of apoptotic and necrotic cells were measured based on morphological changes consistent with apoptotic or necrotic cell death by TEM at 200x or 500-1500X magnification.
(DOCX)

**S2 Table. Statistical analyses of transmission electron microscopy data on T cell death following immune activation.** Fisher's exact test of the 2x2 contingency table was used to assess the significance of the proportion differences between apoptosis and necrosis in stimulated T cells between ME/CFS and healthy control at 500-1500X magnification.
(DOCX)

**S3 Table. Statistical analyses of transmission electron microscopy data on T cell death following immune activation within each pair (identical twin or unrelated pair).** Fisher's exact test of the 2x2 contingency table was used to assess the significance of the proportion differences between apoptosis and necrosis in stimulated T cells between the identical twins

discordant with moderate form of ME/CFS or unrelated participants discordant with extreme form of ME/CFS. This was measured by TEM at 200x or 500-1500X magnification.
(DOCX)

**S4 Table. Statistical analyses of transmission electron microscopy data on cell death in unstimulated and stimulated isolated PBMC subpopulation lacking T cells.** Fisher's exact test of the 2x2 contingency table to assess the significance of the proportion differences between apoptosis and necrosis in PBMC subpopulation lacking T cells. Cells were incubated in the presence or absence of 100nM PMA for 12 h and number of apoptotic and necrotic cells were measured in unstimulated and PMA-stimulated cells by TEM at 200-1500X magnification, based on morphological changes consistent with apoptotic or necrotic cell death.
(DOCX)

**S5 Table. Quantitative analysis of transmission electron microscopy data on mitochondrial ultrastructural abnormalities in stimulated T cells from ME/CFS patients or healthy controls.** Isolated T cells were stimulated with anti-CD3/CD28 beads for 12 h. Mitochondria were counted per cell and assessed for morphological changes (normal, vesicular/compartmentalized or swollen). MT with vesicular/ compartmentalized and swollen morphologies were considered abnormal. This was measured by TEM at 300-2500x magnification.
(DOCX)

**S6 Table. Statistical analyses of transmission electron microscopy data on mitochondrial ultrastructural abnormalities in stimulated T cells.** Fisher's exact test of the 2x2 contingency table was used to assess the significance of the proportion differences between variations in mitochondrial morphology (normal, vesicular/compartmentalized or swollen), with abnormal mitochondria being the sum of vesicular/compartmentalized and swollen per sample. This was performed in stimulated T cells isolated from an identical twin discordant with moderate ME/CFS and a pair of unrelated participants discordant with extreme form of ME/CFS.
(DOCX)

**S7 Table. Quantitative analysis of transmission electron microscopy data on mitochondrial ultrastructural abnormalities in activated T cells.** The severity in mitochondrial dysfunction was assessed via calculating the percentage of cells per each group carrying more than 3 swollen or 6 abnormal mitochondria per single cell.
(DOCX)

**S8 Table. Statistical analyses of transmission electron microscopy data on the severity of mitochondrial ultrastructural abnormalities in stimulated T cells.** Fisher's exact test of the 2x2 contingency table was used to assess the significance of the proportion difference in severity in mitochondrial dysfunction based on the presence of $\geq 3$ swollen or 6 abnormal mitochondria per single cell.
(DOCX)

**S9 Table. Quantitative analysis of transmission electron microscopy data on intracellular and extracellular lipid droplets-like vesicles.** Intracellular and extracellular lipid droplets-like vesicles were counted in unstimulated and stimulated PBMC subpopulation from TEM micrographs at 300 to 800x.
(DOCX)

**S10 Table. Statistical analyses of transmission electron microscopy data on intracellular and extracellular lipid droplets-like vesicles.** Fisher's exact test of the 2x2 contingency table was used to assess the significance of the proportion differences in intracellular and

extracellular lipid droplet-like vesicles in stimulated PBMC subpopulation lacking T cells between unrelated extremely severe ME/CFS patient and unrelated healthy control.
(DOCX)

**S11 Table. Quantitative analysis of transmission electron microscopy data on giant platelet, platelet clump and giant rosette like-platelet aggregate.** Giant platelet, platelet clump and giant rosette like-platelet aggregate were counted in stimulated and unstimulated PBMC subpopulation from TEM micrographs.
(DOCX)

**S12 Table. Statistical analyses of transmission electron microscopy data on giant platelet, platelet clump and giant rosette-like platelet aggregate.** Fisher's exact test of the 2x2 contingency table to assess the significance of the proportion differences between giant platelet, platelet clump and giant rosette-like platelet aggregate counts in unstimulated and stimulated PBMC subpopulation between ME/CFS and healthy controls.
(DOCX)

**S13 Table. Statistical analyses of transmission electron microscopy data on giant platelet, platelet clump and giant rosette-like platelet aggregate in the unrelated pair.** Fisher's exact test of the 2x2 contingency table to assess the significance of the proportion differences between giant platelet, platelet clump and giant rosette-like platelet aggregate counts in unstimulated and stimulated PBMC subpopulation between unrelated extremely severe ME/CFS and unrelated healthy control.
(DOCX)

**S1 Fig. TEM micrograph showing phagocytosis, apoptosis, and autophagy in unstimulated and stimulated PBMCs.** A) TEM image displaying immune synapse formation between a few PBMCs. Note the cell in the center has phagocytosed a platelet, (inset) shows how the plasma membrane of cell forms a pocket to engulf the platelet. B) A phagosome, which contains a large electron dense particle, 1 μm in diameter. C) Two apoptotic cells (arrow), with the apoptotic cell on the left displaying a large apoptotic body (open arrow). D) Cell undergoing autophagic cell death, demonstrating autophagic-like vacuoles, filled with the amorphous materials (arrowheads), the membranous inclusions (open arrowheads) or the organelles (circle) at the various stages of degradation.
(TIF)

**S2 Fig. Electron micrographs identify the presence of "giant lipid droplets" in stimulated T cells, as well as unstimulated and stimulated PBMC lacking T cells.** (A, C, E) Representative images of lipid droplet-like vesicles (arrow) in stimulated T cells from unrelated extremely severe ME/CFS patient (UCFS-T+Act). (B, D, F) Representative images of intracellular (arrow) and extracellular (open arrow) "giant lipid droplet-like vesicles" in stimulated PBMC lacking T cells from unrelated extremely severe ME/CFS (UCFS-P-T+Act). Note the difference in morphology and electron density of these "giant lipid droplet-like vesicles" in comparison to the unstimulated PBMC lacking T (P-T) cells and stimulated T cells (T+Act). (B) A giant lipid droplet-like vesicle compressing the nucleus of the cell (arrowhead). (G, H) Only one unstimulated PBMC lacking T cell from both the twin healthy control (THC-P-T) and the unrelated healthy control (UHC-P-T) contained a "lipid droplet- like vesicle".
(TIF)

**S3 Fig. Transmission electron micrographs of microvilli and extracellular organelles present in unstimulated and stimulated PBMC subpopulations.** A) Interdigitating microvilli of stimulated T cells generating immune synapses. Microvilli can penetrate the glycocalyx,

creating close-contact zones necessarily for immunological synapse function, and are thought to enable message transfer between cells, survey surfaces of antigen-presenting cells and carry T cell receptors. B) Displaying an immune cell along with several platelets, (inset) either representing PBMC microvilli formation near a platelet or platelet microparticles formation near a PBMC to form immune synapse. C) PBMC cell microvilli form budding vesicles constituting immunological synaptosomes, (inset) microvilli reaching out to potentially some platelet-derived microvesicles (PMVs) (open arrow). Microvesicles are important in cell–cell communication and cell differentiation. D) Exhibiting an immunological synapse between a PBMC and two red blood cells. Small particles can be seen in immune synapse junction, (inset) microvilli forming from the immune cell surface. E) The immune synapse between two immune cells, (inset) immune synapse is a site of intense vesicular trafficking, which can be seen as small electron dense particles around microvesicle-like structures (arrowhead) in immune synapse junction. F) Immune synapses (arrow) present between four immune cells. Microvesicles (open arrowhead) and small electron dense particles (arrowhead) can also be seen.
(TIF)

**S4 Fig. Transmission electron micrographs of typical intracellular organelles present in unstimulated and stimulated PBMCs.** A) Golgi apparatus (G) near the nucleus and a few mitochondria (MT) can be seen, (Inset) ultrastructure of Golgi apparatus showing cisternae (white arrow) and large vesicles (white arrowhead). B) Endoplasmic reticulum (open arrow), (insert) ultrastructure of endoplasmic reticulum. C) Displaying a cluster of lysosome like-vesicles (Ly), (inset) typical ultrastructure of a lysosome showing spherical membrane bound organelles with an electron-dense cores indicative of high protein concentration. D) Autophagosome like-vesicle (Au) containing cytoplasmic materials.
(TIF)

**S5 Fig. Transmission electron micrographs of several intracellular organelles present in unstimulated and stimulated PBMCs.** A) Vesicles (V) and multilamellar bodies (MLB) (arrows), which are membrane bound lysosomal vacuoles, (inset) ultrastructure of a MLB showing a membrane bound organelle containing concentric membrane layers. B) Multivesicular bodies (MVB) (open arrows), a particular type of endosome that contains membrane-bound intraluminal vesicles, (inset) ultrastructure of MVB with luminal vesicles (open arrowhead). C) Centriole (arrowhead) in a longitudinal orientation. D) Centriole cross section showing distinct microtubule triplet organization.
(TIF)

## Acknowledgments

We would like to acknowledge ME/CFS patients, their caregivers, and medical team, and control subjects for their generous participation in this study. We would like to acknowledge Dr. Holden Terry Maecker, Director of the Human Immune Monitoring Center at Stanford University for his support. The TEM procedure was supported, in part, by ARRA Award Number 1S10RR026780-01 from the National Center for Research Resources (NCRR). Its contents are solely the responsibility of the authors and do not necessarily represent the official views of the NCRR or the National Institutes of Health. We thank members of the Stanford Center for Genomics and Personalized Medicine, Stanford Genome Technology Center, Lucia Ramirez, Ada Chen, Teodoro Mappala from Stanford Dept of Genetics, Cort Johnson (the founder and director of Health Rising), Dr. Susan Meehan, Alexander Honkala from Stanford Healthcare Innovation Lab and Kimberly Hicks and Linda Tannenbaum from OMF.

## Author Contributions

**Conceptualization:** Fereshteh Jahanbani.

**Data curation:** Fereshteh Jahanbani, Rajan D. Maynard.

**Formal analysis:** Fereshteh Jahanbani.

**Funding acquisition:** Ronald W. Davis, Michael P. Snyder.

**Investigation:** Fereshteh Jahanbani.

**Methodology:** Fereshteh Jahanbani, Justin Cyril Sing, Shaghayegh Jahanbani, John J. Perrino, Damek V. Spacek.

**Project administration:** Fereshteh Jahanbani.

**Resources:** Michael P. Snyder.

**Supervision:** Fereshteh Jahanbani, Michael P. Snyder.

**Visualization:** Fereshteh Jahanbani, Rajan D. Maynard, John J. Perrino.

**Writing – original draft:** Fereshteh Jahanbani.

**Writing – review & editing:** Fereshteh Jahanbani, Rajan D. Maynard, Justin Cyril Sing, Michael P. Snyder.

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
