## [Decision Letter · Decision Letter 0]

23 Mar 2022

PONE-D-22-01617Phenotypic characteristics of peripheral immune cells of Myalgic encephalomyelitis/chronic fatigue syndrome via Transmission Electron Microscopy: a pilot studyPLOS ONE

Dear Dr. Jahaniani Kenari,

Thank you for submitting your manuscript to PLOS ONE. We apologize for the delay in responding but we were beset with problems concerning reviewers and Journal administration. Because of the detailed and constructive comments offered by the expert reviewer, however, a thorough review of the manuscript has been provided. After careful consideration, we feel that it has merit but does not fully meet PLOS ONE’s publication criteria as it currently stands. Therefore, we invite you to submit a revised version of the manuscript that addresses the points raised during the review process.

The concerns voiced by the reviewer mainly involve text changes. However, the issue of patient sex in the study must be addressed and the study needs to be representative of both genders for the reasons articulated by the reviewer. When re-submitting the revised manuscript, please also include a detailed point-by-point description of the steps taken to address all reviewer comments as shown below.

We look forward to receiving your revised manuscript.

Kind regards,

Christopher T. Beh, PhD

Academic Editor

PLOS ONE

Journal Requirements:

3.We note that you have indicated that data from this study are available upon request. PLOS only allows data to be available upon request if there are legal or ethical restrictions on sharing data publicly. For more information on unacceptable data access restrictions, please see http://journals.plos.org/plosone/s/data-availability#loc-unacceptable-data-access-restrictions. 

Reviewers' comments:

Reviewer's Responses to Questions

**Comments to the Author**

1. Is the manuscript technically sound, and do the data support the conclusions?

Reviewer #1: Yes

2. Has the statistical analysis been performed appropriately and rigorously? 

Reviewer #1: Yes

3. Have the authors made all data underlying the findings in their manuscript fully available?

Reviewer #1: Yes

4. Is the manuscript presented in an intelligible fashion and written in standard English?

Reviewer #1: Yes

5. Review Comments to the Author

Reviewer #1: This manuscript is worthwhile even though it is a curious combination of an extensive review on mitochondrial morphology in immune cells and ME/CFS along with a case report on two pairs of individuals discordant for ME/CFS. Both are valuable. Despite the authors citing a rather astounding 199 references, they have overlooked some key references that not only need to be added, but also discussed. Because of the “review-like” nature of the manuscript, I have paid particular attention to appropriate citation. I don’t believe any new experiments are required; instead there are important changes needed to the writing.

Foremost among the missing citations is a paper from their own university that examined PBMCs from ME/CFS patients by TEM. https://pubmed.ncbi.nlm.nih.gov/27747291/ Likely they did not notice this one because of its title. The authors cannot claim their work is the first on TEM of PBMCS in the disease so need to eliminate such statements. But they also need to discuss why their results differ from the 2016 work, which is not as strong as the current manuscript, given that the prior work used unfractionated total PBMCs and it does not have the detailed quantitative comparisons found in this manuscript. I suspect the discrepancy might be related to the earlier work using frozen cells but there are other differences as well. The TEM measurements of mitochondrial size are consistent with the lack of differentiation of mass between patients and controls that were observed with an unmentioned fluorescent microscopy approach https://pubmed.ncbi.nlm.nih.gov/31830003/.

The authors appear to omit information about the sex of the 4 individuals, except for a perhaps accidental mention of “his” caregivers. So the severe patient and control must be male, but what sex are the twins? There is abundant evidence of sex differences, including in lipid metabolism, as well as the female predominance of the disease. For example, https://pubmed.ncbi.nlm.nih.gov/34454515/ , which found sex differences is not cited. There are several other papers describing lipid/fatty acid disturbances in ME/CFS that are not cited. https://pubmed.ncbi.nlm.nih.gov/31947545/, https://www.ncbi.nlm.nih.gov/labs/pmc/articles/PMC5365380/ . The Naviaux PNAS paper, which also documented sex differences, is cited but may need more discussion in this context. It is important to know the sex of all 4 individuals to see whether differences—such as lipid accumulation—are consistent with the literature.

A paper of one of the co-authors that was not cited is https://pubmed.ncbi.nlm.nih.gov/34682970/ which showed abnormal elevated cholesterol/HDL ratio in severe ME/CFS patients. Surely the chlolesterol/LDL/HDL ratio and triglycerides, as they are standard clinical tests, must be known for at least the patients, if not the controls, and could be mentioned. Does the severe patient have an abnormal ratio given the finding of lipid droplets? Actually, comparing such levels in identical twins discordant for the disease could be interesting.

It puzzles me when mention is made that $17 to $24 billion dollars is the cost of ME/CFS to society, when the original paper (not cited, https://dynamic-med.biomedcentral.com/articles/10.1186/1476-5918-7-6) was written in 2008. What is the cost in today’s dollars?

This sentence needs to be changed: “Behan et al. who also reported a substantial increase in mitochondrial abnormalities in the muscle cells in post-viral fatigue syndrome patients, a term the authors preferred over ME/CFS for describing the condition.” The reason they did not prefer ME/CFS is that such a term didn’t exist in 1990 when they wrote the article. Chronic fatigue syndrome as a name was not invented until 1988 and was not in use in the UK and Europe at that time and combining ME and CFS was never done. Instead, back then it was well known that ME was a post-viral syndrome.

Subsequent to this statement: The role of exosomes in ME/CFS etiology has just begun to be explored” the authors describe only one paper and omit https://pubmed.ncbi.nlm.nih.gov/29696075/, https://pubmed.ncbi.nlm.nih.gov/31759091/, https://pubmed.ncbi.nlm.nih.gov/33046133/,

and https://pubmed.ncbi.nlm.nih.gov/32034172/.

6. PLOS authors have the option to publish the peer review history of their article (what does this mean?). If published, this will include your full peer review and any attached files.

Reviewer #1: No

---

## [Author Response · Author response to Decision Letter 0]

22 Jul 2022

Dear Dr. Christopher T. Beh, 

Thank you very much for having considered our manuscript “Phenotypic characteristics of peripheral immune cells of Myalgic encephalomyelitis/chronic fatigue syndrome via Transmission Electron Microscopy: a pilot study”, PONE-D-22-01617, by Jahanbani and colleagues. We would like to thank you and reviewer for the positive review, the insightful comments and helpful suggestions, which helped improve the manuscript. We fundamentally agree with all the comments made by you and the reviewer and made every effort to address them accordingly. The revisions are highlighted in green in 'Revised Manuscript with Track Changes'. We will also submit an unmarked version of our revised manuscript without tracked changes as a separate file labeled 'Manuscript'.

Our detailed, point-by-point responses to the editorial and reviewer comments are below:

• Text highlighted in grey indicates responses to the suggestion of Reviewer #1, 

• Text colored in orange indicates responses to Editor’s suggestions, 

• Text colored in blue indicates responses made to comments on Journal Requirements for submitting the revised manuscript.

Two versions of the manuscript are enclosed, one where all the changes have been highlighted in green, and another version without any marks. We hope our manuscript after careful revisions meet your high standards and is suitable for publication.

Best regards,

Fereshteh Jahanbani 

Response to Editor:

Comment 1: After careful consideration, we feel that it has merit but does not fully meet PLOS ONE’s publication criteria as it currently stands. Therefore, we invite you to submit a revised version of the manuscript that addresses the points raised during the review process.

Response: We wish to express our appreciation to the editor for considering our work and inviting us to submit the revised manuscript, as well as for the insightful comments, which have helped us significantly improve the paper.

Comment 2: The concerns voiced by the reviewer mainly involve text changes. However, the issue of patient sex in the study must be addressed and the study needs to be representative of both genders for the reasons articulated by the reviewer. 

Response: Thank you for your comment. We understand the reviewer’s comments on the importance of discussing sex-dependent lipid disturbances in ME/CFS and have edited our text accordingly. We have added gender information and provided a detailed discussion on current view of lipid abnormality in ME/CFS pathophysiology. Despite some inconsistency, abnormalities in the lipid profile have been reported in both male and female with ME/CFS and our data further support these findings. We agree that more studies are needed to better understand lipid abnormality in ME/CFS patients, and their association with disease etiology, onset, severity, prognosis and comorbidity and how it might vary between patients based on sex, age and disease duration. 

Comment 3: When re-submitting the revised manuscript, please also include a detailed point-by-point description of the steps taken to address all reviewer comments as shown below.

Response: We very much appreciate the thoughtful comments from both editor and reviewer. We have considered all the insightful comments and helpful suggestions proposed by the reviewer. In any case, we are open to any further comments. In our rebuttal letter we highlighted our comments in three sections as 'Response to Editor', 'Comment on Journal Requirements’ and 'Response to Reviewer', addressing a detailed point-by-point description of the step taken to address the comments. We also included a marked-up copy of our manuscript that highlights changes made to the original version in green (labeled as 'Revised Manuscript with Track Changes') and an unmarked version of our revised paper without tracked changes (labeled as 'Manuscript').

Comment on Journal Requirements:

Comment 1: Please ensure that your manuscript meets PLOS ONE's style requirements, including those for file naming. 

We apologize for not formatting the manuscript properly based PLOS ONE's required style for both the text and the files’ name. We have now formatted our manuscript according to PLOS ONE's style requirements (such as using level 1 heading for all major sections) and naming supplementary figures properly (e.g., Fig_S1.tif). In our previous version we provided all supplementary tables in a word document, named ‘Supplement ‘. We have now created separate file for each supplementary table (e.g. Table S1, Table S2 and Table S3) and modified our in-text citations for our supplementary tables following PLOS ONE guidelines for in text-citations.

Comment 2: We note that the grant information you provided in the ‘Funding Information’ and ‘Financial Disclosure’ sections do not match. 

Response: We apologize for this mistake and provided the correct grant number used for this study in the ‘Funding Information’ section. 

Comment 3: We note that you have indicated that data from this study are available upon request. 

TEM images for this study are publicly available from the Stanford Digital Repository (https://purl.stanford.edu/zm622tr7008). The de-identified FASTQ files for 4 whole exome sequencing for this study are also publicly available from the Stanford Digital Repository (https://purl.stanford.edu/jd768nw9509).

Comment 4: Please include a separate caption for each figure in your manuscript.

We have ensured that a separate caption for each figure has been provided in the revised manuscript.

Comment 5: Please include captions for your Supporting Information files at the end of your manuscript, and update any in-text citations to match accordingly. 

We have provided captions for our ‘Supporting Information' files at the end of the manuscript.

Comment 6: Please review your reference list to ensure that it is complete and correct. 

We have checked all included articles in the revised manuscript. We increased the number of citations in the manuscript and provided additional references in response to reviewer’s helpful suggestions and updated our citation format based on PLOS ONE guidelines.

Response to Reviewer:

Reviewer Comment 1: This manuscript is worthwhile even though it is a curious combination of an extensive review on mitochondrial morphology in immune cells and ME/CFS along with a case report on two pairs of individuals discordant for ME/CFS. Both are valuable. Despite the authors citing a rather astounding 199 references, they have overlooked some key references that not only need to be added, but also discussed. Because of the “review-like” nature of the manuscript, I have paid particular attention to appropriate citation. I don’t believe any new experiments are required; instead there are important changes needed to the writing.

Reply: We would like to express our appreciation to the reviewer for the comprehensive and positive review of our manuscript, constructive remarks, and pointing out additional citations that merits being included in our manuscript. We have taken the comments on board to improve and clarify the manuscript. Please find below a detailed point-by-point response to all comments (reviewers’ comments in black, and our responses highlighted in grey.

Reviewer Comment 2: Foremost among the missing citations is a paper from their own university that examined PBMCs from ME/CFS patients by TEM. https://pubmed.ncbi.nlm.nih.gov/27747291/ Likely they did not notice this one because of its title. The authors cannot claim their work is the first on TEM of PBMCS in the disease so need to eliminate such statements. But they also need to discuss why their results differ from the 2016 work, which is not as strong as the current manuscript, given that the prior work used unfractionated total PBMCs and it does not have the detailed quantitative comparisons found in this manuscript. I suspect the discrepancy might be related to the earlier work using frozen cells but there are other differences as well.

Reply: We apologize for these omissions and have added the recommended reference to our revised manuscript, as Ref 40. We eliminated our claim in the text (please see page number 5 of the revised manuscript). The reviewer is correct that study design between our work and the 2016 study is very different, which could simply explain inconsistency in findings. We discussed the study designs and possible reasons that might lead to different findings between our work and the 2016 study (please see page number 35 of the revised manuscript). We provided additional citations supporting the discussion (please see Ref No 152-155).

Reviewer Comment 3: The TEM measurements of mitochondrial size are consistent with the lack of differentiation of mass between patients and controls that were observed with an unmentioned fluorescent microscopy approach https://pubmed.ncbi.nlm.nih.gov/31830003/.

We apologize for these omissions and have added the recommended reference to our revised manuscript, as Ref 172 and 173.

Reviewer Comment 4: The authors appear to omit information about the sex of the 4 individuals, except for a perhaps accidental mention of “his” caregivers. So the severe patient and control must be male, but what sex are the twins? 

Reply: We would like to apologize for omitting the information and included the gender in the method section under Participant (please see page number 7 of the revised manuscript) as well as Table 1. Participants sample IDs, used for the TEM study (please see page number 8 of the revised manuscript).

Reviewer Comment 5: There is abundant evidence of sex differences, including in lipid metabolism, as well as the female predominance of the disease. For example, https://pubmed.ncbi.nlm.nih.gov/34454515/, which found sex differences is not cited. 

Reply: We agree with the reviewer and have revised this section in the discussion according to the comments to highlight the significance of the sex-dependent lipid disturbances in ME/CFS (please see page 37 of our revised manuscript). We also like to thank the reviewer for pointing out the citation and have included it in our revised manuscript as Ref 174.

Reviewer Comment 6: There are several other papers describing lipid/fatty acid disturbances in ME/CFS that are not cited. https://pubmed.ncbi.nlm.nih.gov/31947545/, https://www.ncbi.nlm.nih.gov/labs/pmc/articles/PMC5365380/). 

Reply: We appreciate the suggestion regarding the incorporation of the above reference, which permitted us to include aspects that, despite their relevance, were not cited in the original version (please see Ref 172 and 173 in our revised manuscript). 

Reviewer Comment 7: The Naviaux PNAS paper, which also documented sex differences, is cited but may need more discussion in this context. It is important to know the sex of all 4 individuals to see whether differences—such as lipid accumulation—are consistent with the literature.

Reply: We agree with the reviewer that we didn’t provide enough details on sex-specific lipid dysregulation in ME/CFS. In our revised manuscript, we now point this out in the discussion and provided brief summary of sex-dependence lipid/fatty acid disturbances in ME/CFS (please see page 37-38 of the revised manuscript).

Reviewer Comment 8: A paper of one of the co-authors that was not cited is https://pubmed.ncbi.nlm.nih.gov/34682970/ which showed abnormal elevated cholesterol/HDL ratio in severe ME/CFS patients. Surely the chlolesterol/LDL/HDL ratio and triglycerides, as they are standard clinical tests, must be known for at least the patients, if not the controls, and could be mentioned. Does the severe patient have an abnormal ratio given the finding of lipid droplets? Actually, comparing such levels in identical twins discordant for the disease could be interesting.

Reply: We thank reviewer for bringing up the Davis R. et al work. We added the citation to our manuscript as Ref 175. We obtained the clinical data from three of the participants, including the twin with moderate ME/CFS, the extremely severe ME/CFS patient and the unrelated male healthy control. Unfortunately, the healthy twin didn’t have any blood test done within last 6 years. Both patients showed increases in low-density lipoprotein, decrease in HDL and elevated cholesterol/HDL ratio, while the unrelated healthy control had normal values for these features.

Reviewer Comment 9: It puzzles me when mention is made that $17 to $24 billion dollars is the cost of ME/CFS to society, when the original paper (not cited, https://dynamic-med.biomedcentral.com/articles/10.1186/1476-5918-7-6) was written in 2008. What is the cost in today’s dollars?

Reply: We highly agree with the reviewer on the urgent need to further study ME/CFS economic burden and compare the findings with the results from the original paper published in 2008. We have revised our manuscript and included more recent citations (page 3-4 and Ref 9-15 of our revised manuscript), which indicate a dramatic increase in both ME/CFS prevalence and economic impact due to direct and indirect costs.

Reviewer Comment 10: This sentence needs to be changed: “Behan et al. who also reported a substantial increase in mitochondrial abnormalities in the muscle cells in post-viral fatigue syndrome patients, a term the authors preferred over ME/CFS for describing the condition.” The reason they did not prefer ME/CFS is that such a term didn’t exist in 1990 when they wrote the article. Chronic fatigue syndrome as a name was not invented until 1988 and was not in use in the UK and Europe at that time and combining ME and CFS was never done. Instead, back then it was well known that ME was a post-viral syndrome.

Reply: We apologize for the misunderstanding and changed this sentence accordingly (page 34 of the revised manuscript). 

Reviewer Comment 11: Subsequent to this statement: The role of exosomes in ME/CFS etiology has just begun to be explored” the authors describe only one paper and omit https://pubmed.ncbi.nlm.nih.gov/29696075/, https://pubmed.ncbi.nlm.nih.gov/31759091/, https://pubmed.ncbi.nlm.nih.gov/33046133/,

and https://pubmed.ncbi.nlm.nih.gov/32034172/.

Reply: We appreciate the reviewer insightful comments and added the recommend citation to in the manuscript (Please see references 146, 200–202).

---

## [Editor Report · Decision Letter 1]

26 Jul 2022

Phenotypic characteristics of peripheral immune cells of Myalgic encephalomyelitis/chronic fatigue syndrome via Transmission Electron Microscopy: a pilot study

PONE-D-22-01617R1

Dear Dr. Jahaniani Kenari,

We’re pleased to inform you that your manuscript has been judged scientifically suitable for publication and will be formally accepted for publication once it meets all outstanding technical requirements.

Kind regards,

Christopher T. Beh, PhD

Academic Editor

PLOS ONE

Additional Editor Comments (optional):

The authors have addressed all concerns raised by the expert reviewer and the article is much improved. The paper should be of interest to those studying the cell biology of Myalgic encephalomyelitis/chronic fatigue syndrome.
---

## [Editor Report · Acceptance letter]

29 Jul 2022

PONE-D-22-01617R1 

Phenotypic characteristics of peripheral immune cells of Myalgic encephalomyelitis/chronic fatigue syndrome via Transmission Electron Microscopy: a pilot study 

Dear Dr. Jahanbani:

I'm pleased to inform you that your manuscript has been deemed suitable for publication in PLOS ONE. Congratulations! Your manuscript is now with our production department. 

Kind regards, 

on behalf of

Dr. Christopher T. Beh 

Academic Editor

PLOS ONE